# Chemical Reactivities of *ortho*-Quinones Produced in Living Organisms: Fate of Quinonoid Products Formed by Tyrosinase and Phenoloxidase Action on Phenols and Catechols

**DOI:** 10.3390/ijms21176080

**Published:** 2020-08-24

**Authors:** Shosuke Ito, Manickam Sugumaran, Kazumasa Wakamatsu

**Affiliations:** 1Department of Chemistry, Fujita Health University School of Medical Sciences, 1-98 Dengakugakubo, Kutsukake-cho, Toyoake, Aichi 470-1192, Japan; 2Department of Biology, University of Massachusetts, Boston, MA 02125, USA; Manickam.Sugumaran@umb.edu

**Keywords:** *o*-quinones, quinone methides, phenols, catechols, tyrosinase, thiols, amines, dopaquinone, sclerotization, melanization

## Abstract

Tyrosinase catalyzes the oxidation of phenols and catechols (*o*-diphenols) to *o*-quinones. The reactivities of *o*-quinones thus generated are responsible for oxidative browning of plant products, sclerotization of insect cuticle, defense reaction in arthropods, tunichrome biochemistry in tunicates, production of mussel glue, and most importantly melanin biosynthesis in all organisms. These reactions also form a set of major reactions that are of nonenzymatic origin in nature. In this review, we summarized the chemical fates of *o*-quinones. Many of the reactions of *o*-quinones proceed extremely fast with a half-life of less than a second. As a result, the corresponding quinone production can only be detected through rapid scanning spectrophotometry. Michael-1,6-addition with thiols, intramolecular cyclization reaction with side chain amino groups, and the redox regeneration to original catechol represent some of the fast reactions exhibited by *o*-quinones, while, nucleophilic addition of carboxyl group, alcoholic group, and water are mostly slow reactions. A variety of catecholamines also exhibit side chain desaturation through tautomeric quinone methide formation. Therefore, quinone methide tautomers also play a pivotal role in the fate of numerous *o*-quinones. Armed with such wide and dangerous reactivity, *o*-quinones are capable of modifying the structure of important cellular components especially proteins and DNA and causing severe cytotoxicity and carcinogenic effects. The reactivities of different *o*-quinones involved in these processes along with special emphasis on mechanism of melanogenesis are discussed.

## 1. Introduction

*ortho*-Quinones are ubiquitous in nature. Plants especially produce a wide variety of *o*-quinones as secondary metabolites. It is not possible to cover all naturally occurring quinones and their reactions in a short review. Therefore, only quinones arising from catecholamine derivatives and few other related molecules will be considered in this review. Unlike the related *p*-quinones that are reasonably stable towards isolation and characterization, several of the *o*-quinones are extremely reactive and difficult to isolate. Often, they are present in the transient form in a reaction thus even avoiding detection. A most important and prominent example of the *o*-quinones is dopaquinone that plays pivotal roles in melanogenesis, the process of production of eumelanin and pheomelanin [1]. Due to its inherent high reactivity dopaquinone can easily undergo both Michael 1,6- and 1,4-additions with nucleophiles such as thiols and amines. These reactions lead to the production of melanin pigment found in the skin, hair, and eyes of all animals. The reactivity of the related dopaminequinone produces neuromelanin in the brain [2,3]. Protein dopaquinones formed in mussel proteins are responsible for gluing of mussels to rocks and other hard surfaces [4]. The defense reaction exhibited by tunichromes found in the blood of tunicates are due to the reactivities of *o*-quinones [5]. In insects, catecholamine derivates such as *N*-acetyldopamine and *N*-β-alanyldopamine form key components of hardened exoskeleton that protects the soft bodied animals [6]. In plants and fungi, oxidative browning reactions that are part of their defense mechanism against invading insects, is caused by the reactivities of enzymatically generated *o*-quinones. While some of the reactions are beneficial to the organisms, several of the reactions of *o*-quinones especially binding to macromolecules such as proteins and DNA through the sulfhydryl and/or amino groups, results in deteriorating biological consequences such as cytotoxicity and carcinogenesis. Therefore, it is important to understand the biological activities of these transient compounds. *o*-Quinones are mostly produced in biological systems by the oxidation of the corresponding catechols (*o*-diphenols) by two electron removal. However, tyrosinase and related phenoloxidases present in mammals, plants, fungi, and insects are capable of oxidizing simple phenols to the corresponding quinones [7] (Figure 1). This review summarizes briefly the function of tyrosinase and the chemical reactivity of *o*-quinones with various nucleophiles. Special emphasis is given to the mechanisms of melanogenesis. At this point, we want to point out that practically all biological reactions are enzyme-catalyzed, with the exception of ribozymes, which are RNA catalysts. Without enzymatic intervention, some reactions can proceed, but for them to be useful for the biological system, enzymatic intervention is absolutely necessary. An example is hydration of carbon dioxide to carbonic acid which can proceed quite rapidly without the need for an enzyme, but in biological systems, carbonic anhydrase makes this hydration go faster. However, a very small set of reactions, form a group of nonenzymatic reactions that often occur in biological systems. Glycation of hemoglobin is one such example. While it is a useful index to determine the seriousness of diabetic conditions, the glycation is entirely of nonenzymatic origin. In the case of *o*-quinone chemistry, once these reactive intermediates are generated by enzyme assisted reactions, the rest of the reactions seem to proceed nonenzymatically without the need for any enzymatic assistance. These nonenzymatic reactions are absolutely essential for parts of melanin biosynthesis, insect cuticular sclerotization, innate immunity in invertebrate animals where encapsulation and melanization of foreign objects occur, mussel glue formation (1,4–6). Therefore, it is important to investigate the fate of *o*-quinones in biological systems, which forms the main goal of this review.

## 2. *o*-Quinone Formation from Phenols and Catechols by Tyrosinase

Tyrosinase (EC 1.14.18.1) is a rate-limiting enzyme for the biosynthesis of melanin pigments and is present widespread in nature, ranging from microorganisms, plants, insects to mammals. The primary function of tyrosinase is the production of *o*-quinones from both phenols and catechols. This enzyme exhibits two different activities. Often described in literature as monophenol monooxygenase (or monophenolase) activity is assumed to convert monophenols to *o*-diphenols, although in reality the diphenol is produced by the reduction of *o*-quinone primary product. Thus, this activity converts monophenols to *o*-quinones first. The second activity is the oxidation of *o*-diphenols to *o*-quinones and is referred to as catechol oxidase or diphenolase activity (Figure 1). Oxidases such as peroxidase, laccase, and polyphenol oxidase are also able to catalyze the oxidation of catechols to *o*-quinones. Typically, many biological oxidants produce *o*-quinones from catechols but not from phenols. Thus, only tyrosinase is capable of exhibiting both phenol monooxygenase activity and diphenoloxidase activity. Therefore, the function of tyrosinase is briefly summarized. 

Tyrosinases can be classified into type-3 copper proteins. This class of functional proteins includes tyrosinases, catechol oxidases, hemocyanins (the oxygen-carrying proteins of arthropods and mollusks), and arthropod phenol oxidases. The active site of these proteins consists of two copper atoms (identified as CuA and CuB), each coordinated by three histidine residues (Figure 2). Several tyrosinases have been isolated and their crystalline structures elucidated. For example, the tyrosinase from *Agaricus bisporus* is a heterotetramer composed of two heavy chains of about 48 kDa and two light chains of 14 kDa [8]. The first 3D structure study of this protein indicates that three histidine residues each in the heavy chain coordinate to two copper atoms (CuA and CuB) [8]. 

Tyrosinase possesses four distinctly different oxidation states (*deoxy*-, *oxy*-, *met*-, and *deact*-) and understanding their roles and inter-conversions is essential for understanding catalytic activities of tyrosinase [7,9]. Tyrosinase is usually present mostly as a stable *met*-tyrosinase [Cu(II)_2_], in which a hydroxide ion is bound to two Cu(II) atoms. This form of the enzyme does not react with phenols but is able to oxidize catechols to *o*-quinones. During this oxidation process, *met*-tyrosinase [Cu(II)_2_] is reduced to *deoxy*-tyrosinase [Cu(I)_2_]. The latter rapidly reacts with molecular oxygen to form the bound peroxide ion in *oxy*-tyrosinase [Cu(II)_2_-O_2_]. *oxy*-Tyrosinase is the most important oxidizing form of this enzyme and oxidizes phenols to *o*-quinones while it returns reversibly to *deoxy*-tyrosinase. The latter binds to molecular oxygen to regenerate *oxy*-tyrosinase and the oxidation of phenols continues until the substrates phenol and molecular oxygen are depleted. This is the phenol-oxidase cycle.

*oxy*-Tyrosinase is able to oxidize catechols to *o*-quinones and itself is reduced to *met*-tyrosinase. The latter oxidizes the second molecule of catechol to *o*-quinone through the catechol oxidase mechanism and itself is reduced to *deoxy*-tyrosinase (mentioned above). This is the catechol-oxidase cycle. The hydroxyl group of phenols binds to the active site through the CuA, while catechols initially bind to the CuB [10]. Therefore, phenols and catechols act competitively against *oxy*-tyrosinase. Usually *oxy*-tyrosinase exhibits a higher specificity for catechol oxidation than for the corresponding phenol oxygenation reaction. 

As pointed out earlier, it is still generally claimed that tyrosinase first hydroxylates phenols to catechols that subsequently undergo oxidation to *o*-quinones. However, it should be emphasized that in the monooxygenase mechanism (Figure 1), the *o*-quinone is formed directly from the phenol and catechol formation is not an intermediate step [7]. A possible reason for this misunderstanding is the fact that the quinone is reduced by a nonenzymatic redox exchange mechanism thus producing the catechol [11] (see Section 6).

It has been known for years that tyrosinase is slowly inactivated during the oxidation of catechols. The reason for such suicidal inactivation remained unidentified till Land et al. [12,13] pointed out that catechol substrates may sometimes (once in 2000 turnover) be oxidized by *oxy*-tyrosinase through the monooxygenase mechanism and *oxy*-tyrosinase is irreversibly reduced to *deact*-tyrosinase (Figure 2). During this process, one of the two copper atom is reduced to the Cu(0) state and is released from the active center. This proposal is consistent with the observation the inactivation of tyrosinase activity is accompanied by a loss of half the active-center copper atoms [7]. Recently, a similar mechanism was proposed for the inactivation of tyrosinase by resorcinol [14]. 

The catecholic substrate that is formed through redox exchange is able to activate the native form of tyrosinase, *met*-tyrosinase, to the active form, *oxy*-tyrosinase. Evidence in support of this proposal came from the study by Cooksey et al. [11]. *N*,*N*-Dimethyltyramine is oxidized to the corresponding *o*-quinone and undergoes cyclization. The cyclized product, indolium salt, is unable to act as a catecholic (diphenolic) substrate for *met*-tyrosinase due to the presence of the electron-withdrawing Me_2_N^+^ group keeping one of the hydroxyl groups as oxide anion [9] (Figure 3). With this background in mind, let us examine the reactivities of quinonoid products that are formed in the biological processes.

## 3. Reactivity of *o*-Quinones with Small Molecules Leading to Adducts Formation

### 3.1. Chemical Fate of o-Quinones

In this section, chemical fates of *o*-quinones are summarized. Readers are advised to refer also to excellent reviews on related topics with different points of view [6,9,15,16,17,18,19,20,21,22]. 

*o*-Quinones are extremely reactive electrophiles that undergo Michael-1,4-addition reactions with available nucleophilic molecules. However, thiols exhibit a different behavior. The reactions of thiols with quinones are extremely fast–faster than any other nucleophiles. But they also deviate from other nucleophiles in their regiospecificity to the addition site. They add on to the carbon atom next to the *o*-quinone moiety (O3 and O4) at C2- and C5-positions (1,6-Michael addition) and not to the remote C6-position (1,4-Michael addition), the site preferred by the most other nucleophiles. This regiospecificity of thiol addition remains enigma in the chemistry of *o*-quinones. Kishida and Kasai [23] investigated this process through density functional theory-based calculations, showing that the binding of Cys–S^-^ to dopaquinone proceeds with recruiting of Cys–S^-^ to C3–C4 bridge, migration of Cys–S^-^ to C5, proton rearrangement from cysteinyl-NH_3_^+^ to O4, and from C5 to O3. But more studies are needed to get a clear picture. With this exception in mind, the chemical reactivity of *o*-quinones can be classified into four categories based on the appropriate speed of reactions (Figure 4): (1) addition of thiols (through sulfhydryl group) by 1,6-Michael reaction, (2) reduction to the parent catechols through redox exchange, (3) addition of amines (through amino group) by 1,4-Michael reaction or Schiff base formation, and (4) addition of other nucleophiles such as carboxylic acids (through carboxylate anion), phenols, alcohols, and water (in this order of reactivity). The rate of these four sets of the reactions are: Reaction 1 (fast) > Reaction 2 (next fast) >> Reaction 3 (slow) > Reaction 4 (very slow). In Figure 4, the thickness of the arrows corresponds to the fastness of these reactions. In this review, the “fast” reactions complete within about a second, while the “slow” reactions require minutes to hours to complete. The type 4 reactions often do not proceed unless the functional group is present within the same molecule of *o*-quinone. In this last category one can also include slow dimerization and other oligomerization reactions. However, such reactions are so complex that general conclusions cannot be drawn and thus are not discussed in this review.

### 3.2. Reaction of o-Quinones with Thiols

The reactivity of thiols with *o*-quinone is astonishingly very fast. For example, one would expect that a nucleophile which is present internally on the quinone at a suitable position would have the greatest advantage of using proximity effect to exhibit extremely fast intramolecular cyclization as opposed to a nucleophile presented externally to the quinone. However, thiol addition defies such logical explanation and proceeds remarkably faster than even the intramolecular cyclization reactions. An early study, by Tse et al. [24], examined nucleophilic addition rate constants of nucleophiles against electrochemically generated dopaminequinone. At pH 7.4, cysteine and glutathione were found to react faster than the intramolecular cyclization by 2100 and 1400 times, respectively. Lysine was less reactive with a relative rate of 0.47 compared to the cyclization. Ascorbic acid reduced dopaminequinone back to dopamine through redox exchange much faster than the cyclization, but surprisingly, the addition of glutathione was found to proceed a little faster than the reduction reaction. The detailed kinetic study conducted by Jameson et al. [25] also pointed out the extremely fast addition of cysteine to dopaminequinone. At a concentration of 50 µM (physiological concentration in the brain) cysteine reacted 2000-times faster than the intramolecular cyclization. 

The high reactivity of sulfhydryl group in thiols is illustrated by the addition reaction of cysteine to dopaquinone generated in situ by tyrosinase-catalyzed oxidation of dopa [26]. The reaction afforded cysteine adducts in a quantitative yield: 5-*S*-cysteinyldopa (5SCD), 2-*S*-cysteinyldopa (2SCD), 6-*S*-cysteinyldopa (6SCD), and 2,5-*S*, *S*-dicysteinyldopa (2,5DiCD) in yields of 74%, 14%, 1%, and 5%, respectively (Figure 5). As shown in Section 6.1, these cysteinyldopa isomers are further oxidized to give rise to pheomelanin. The facile production of 5-*S*-cysteinyldopa also makes it an excellent biochemical marker for melanoma progression [27,28]. Interestingly, 2,5-*S*,*S*-dicysteinyldopa was found as a reflecting material in the tapetum lucidum (a reflecting later lying behind the retina) of the eyes of the alligator gar *Lepisosteus* [29]. Dopaminequinone reacted similarly with cysteine to afford cysteinyldopamine isomers in comparable yields [30] and dopaquinone reacted similarly with glutathione [31]. Cysteinyldopamine isomers are important precursors of the brain melanin, neuromelanin [32,33]. Regarding the reactivity of various other *o*-quinones with thiols, Cooksey et al. [34] obtained rate constants for reactions of cysteine and glutathione with 17 different *o*-quinones and concluded that the rate constants increase with the electron withdrawing capacities of the substituent groups. This is principally due to the resonance effect, with a smaller but significant contribution attributable to the field effect.

As shown for the reaction of dopaquinone or dopaminequinone with thiols (Figure 5), di-adduct formation is usually a minor pathway. A unique exception is the reactivity of resveratrol quinone (Figure 6). Tyrosinase-catalyzed oxidation of resveratrol, 3,5,4′-trihydroxy-*trans*-stilbene, in the presence of four equivalent *N*-acetylcysteine (NAC) afforded triNAC-resveratrol-catechol and diNAC-resveratrol-catechol in 24% and 22% yields, respectively [35]. Surprisingly, no mono-adduct was isolated. The production of triGSH adduct of resveratrol was also reported by De Lucia et al. [36]. Similarly, chlorogenic acid, a catecholic compound with a conjugated double bond, produced the triGSH adduct along with diGSH and monoGSH adducts showing that the C2 position is most reactive [37] (Figure 6). The reactions of *N*-acetylcysteine with the quinones produced from catechol, 4-methylcatechol and *N*-acetyldopamine, have also been reported [38]. The quinone formed from 1,2-dehydro-*N*-acetyldopamine also trapped *N*-acetylcysteine in a complex reaction yielding multiple products [5].

A unique example of di-thiol adducts of dopa was found in adenochrome, an iron (III)-binding peptide from *Octopus vulgaris* [39]. Acid hydrolysis of desferri-adenochrome afforded a mixture of adenochromine isomers and glycine in a molar ratio of 1:2. Structure elucidation of adenochromines indicated that they consisted of the three possible isomers of di-adducts of 5-thiolhistidine to dopa (Figure 7). Biosynthetic pathway to adenochromines is not well clarified as tyrosinase-catalyzed oxidation of dopa in the presence of 5-thiolhistidine afforded 85% yield of the three possible mono-adducts with only 3% yield of the di-adducts [40].

Methionine which possesses a thioether group also reacts with *o*-quinones but at much less rate compared to the free thiol group of cysteine and related compounds. Vithayathil and his group [41,42,43,44,45] reported that reaction of simple *o*-benzoquinone with *N*-acetylmethionine in acetic acid resulted in the formation of *N*-acetylmethionylcatechol (Figure 8). Subsequently, they were able to show that this reaction is of general occurrence and a number of *o*-quinones could react with methionyl group. They showed that even methionine residues present in ribonuclease were able to react with *o*-benzoquinone through the thioether group [41]. The physiological relevance of this reaction was later demonstrated by Sugumaran and Nelson [46], who reported the production of *N*-acetylmethionine adducts during the tyrosinase-catalyzed oxidation of catechol, 4-methylcatechol, and *N*-acetyldopamine. These authors also proposed that such reactions could be of biological significance during sclerotization insect cuticle, although it was acknowledged by them that insect cuticular proteins have low concentration of sulfur containing amino acids. Moreover, these addition reactions also required large mole excess of *N*-acetylmethionine. Therefore, occurrence of such reactions although theoretically possible, nevertheless is unlikely to witness in insect systems that biological significance of this type of conjugation with proteins is questionable. 

### 3.3. Reaction of o-Quinones with Amines

Next to the sulfhydryl group, the amino group reacts with *o*-quinones at rates that are biologically relevant especially when it is present within the same molecule of *o*-quinone. Three types of reactions have been characterized so far for amines: (1) intramolecular cyclization if the amino group is situated within the molecule at a suitable position, (2) intermolecular nucleophilic addition, and (3) Schiff base formation. The intramolecular cyclization of side chain amino group in dopaquinone and dopaminequinone results in the production of cyclodopa and cyclodopamine. These two reactions form critical steps in eumelanogenesis and neuromelanogenesis (Figure 9) and will be discussed in detail in Section 6. Such unique reactivity is not specific for a particular *o*-quinone alone, but is of general occurrence is demonstrated by the pioneer work by Hawley et al. [47]. Electrochemically generated *o*-quinones from epinephrine and norepinephrine were found to cyclize 140- and 10- times faster than dopaminequinone. Interestingly, the high tendency of epinephrinequinone and norepinephrinequinone to cyclize to form stable aminochromes is related to the lower cytotoxicity of epinephrine and norepinephrine [48]. Reactivity of dopaminequinone was compared with dopaquinone by Li and Christensen [49]. Cyclization of dopaquinone proceeded ca. 10 times faster than dopaminequinone. The faster cyclization of dopaquinone is related to the lower cytotoxicity of dopa as compared to dopamine [48]. Another example of cyclization is presented during the oxidation of 4-*S*-cysteaminylphenol. Mascagna et al. [50] showed that a violet pigment, dihydro-1,4-benzothiazine-6,7-dione (BQ), was produced upon tyrosinase oxidation of 4-*S*-cysteaminylphenol (4SCAP; Figure 9). A seven-membered homologue of BQ can also be produced from 4-*S*-homocysteaminylphenol [51]. 4SCAP is highly cytotoxic to melanoma cells [52], which can be attributed to the tyrosinase-dependent production of BQ [53].

Nucleophilic addition of amines to *o*-quinones has not been extensively examined as compared to thiols. This may be ascribed to the lower reactivity of amines and the instability of the amine adducts produced. Li et al. [54] compared reactivity of small thiols and amines with *o*-quinones. They found the reaction of 4-methyl-*o*-benzoquinone with the amino group of amino acids (Gly, *N*α-acetyl-L-Lys, *N*ε-acetyl-L-Lys, and L-Lys) and the guanidine group of *N*α-acetyl-L-Arg was at least 5 × 10^5^ slower compared to the reaction with the low molecular mass thiols such as cysteine. In addition, they also observed formation of transient reduced amine-quinone adducts followed by reoxidation to the more stable, oxidized form having absorbance near 500 nm (Figure 10). More recently, Li et al., [55] compared the reactivity of different *o*-quinones with lysine and cysteine using cyclic voltammetry. The study indicated the order of reactivity of various quinones as: protocatechuic acid quinone = *o*-benzoquinone > 4-methyl-*o*-benzoquinone = caffeic acid quinone > chlorogenic acid quinone. The reactivity of quinones appear to decrease with steric hinderance of substituents on quinone ring and weakened by enhancing electron cloud density on the quinone ring. Adducts generated by the interaction of 4-methyl-*o*-benzoquinone with amines were identified as amine-quinone adducts connected at the C6 position (Figure 4). The reactions of primary amine with *o*-quinone is very complex and the products suffer further reactions and hence a clear picture is very difficult to obtain. However, the reactions of secondary amine type—especially imidazole with *o*-quinone lead to clean products that have been fully characterized.

In 1982, Sugumaran and Lipke [56] pointed out the involvement of both side chains of lysine and histidine in cross-linking reactions with insect cuticular quinonoid molecules through radioactive experiments. Following this study, Xu et al. [57] reported using model sclerotization studies with *N*-acetyldopaminequinone and *N*-acetylhistidine the formation of 6-*N*- and 2-*N*-imidazole adducts in a ratio of 5:1 (Figure 11). The position of amine substitution was confirmed by NMR studies. Cyclic voltammetry studies with these adducts and parent catechol revealed that they are much more resistant to oxidation in comparison with *N*-acetyldopamine, which explains why these compounds are produced as stable products in contract to the reaction products of primary amines with quinone, which are extremely unstable. The production of 6-*N*-imidazole adduct is consistent with the fact that amines obey predominantly typical Michael-1,4-nucleophilic addition reaction, contrary to the thiol additions. However, the formation of 2-*N*-imidazole adduct and the absence of 5-*N*-imidazole adduct is intriguing as sterically 5-*N*-adduct is a more preferred product than the 2-*N*-adduct yet, it is not reportedly formed in the reaction mixture. More studies are needed to shed light on this inconsistent observation.

Reactions of *o*-quinones with amino group in deoxyribonucleosides have also been reported. Cavalieri et al. [58] examined the reaction of *o*-quinones generated by chemical oxidation of catechol, dopamine, and *N*-acetyldopamine with Ag_2_O or NaIO_4_, and witnessed their addition reaction with adenine (Ade) and deoxyguanosine (dG) (Figure 12). The position of amine addition was confirmed to be C6 by NMR, again in accordance with the Michael-1,4-addtion. These adducts also served as standards for evaluating the binding of DNA to *o*-quinones (see Section 5.2). Interestingly, these adducts were produced at the reduced, catecholic state. Under the same conditions, however, deoxyadenosine, deoxycytidine, and thymidine did not exhibit any reaction with *o*-benzoquinone.

The third type of reaction of *o*-quinone with amine is the Schiff’s base formation. The carbonyl group of quinone and the primary amino groups can easily condense and loose water to form Schiff bases (Figure 4). The quinone imine formation is vital for the biosynthesis of pheomelanin pigment. Cysteinyldopa quinone exhibits intramolecular cyclization and forms a bicyclic quinone imine which is the precursor for all pheomelanin production. More about this will be discussed in Section 6.2. But the reaction does not have to occur only intramolecularly. Even intermolecular addition of amines to *o*-quinones have been witnessed. Such reactions occur during the oxidative transformation of insect cuticular sclerotizing precursor, *N*-β-alanyldopamine. It accounts for the brown coloration of insect cuticle. Mass spectral studies have tentatively identified the production of different colored quinone imine adducts [59].

### 3.4. Reaction of o-Quinones with Carboxylic Acids, Phenols, and Alcohols (Water)

The reaction of quinones with carboxyl groups are sluggish. Thus, the reactions of *o*-quinones with carboxylic acids (carboxylates) would not proceed unless the carboxylate group is present in the side chain of the *o*-quinone (Figure 4). Sugumaran et al. [60] showed that carboxyethyl-*o*-benzoquinone undergoes cyclization to form the six-membered lactone derivative, dihydroesculetin (Figure 13). Interestingly, the lower homologue, carboxymethyl-*o*-benzoquinone (DOPAC quinone), also afforded the five membered lactone ring that tautomerizes to form 2,5,6-trihydroxybenzofuran (Figure 13) along with 3,4-dihydroxymandelic acid (DOMA) and 3,4-dihydroxybenzaldehyde (DHIBAld) formed *via* quinone methide intermediate (see the next section; [61]). Another example is that 1,2-dehydro-*N*-acetyldopa upon oxidation rapidly undergoes intramolecular cyclization [62]. Even though quinone may be the initial product of the reaction, it rapidly and instantaneously isomerized to a tautomeric quinone methide imine amide that exhibited rapid intramolecular cyclization generating dihydroxy coumarin derivative (see the next section; Figure 13).

The 2,5,6-trihydroxybenzofuran formed from the cyclization of carboxymethyl-*o*-benzoquinone suffered further oxidation but characterization of the product turned out to be tough [61]. To overcome this problem, fluorine substituted 3,4-dihydroxyphenylacetic acid was synthesized and its oxidation examined. Fluorine positioned at the cyclization site left during cyclization of the quinonoid product, due to its high electronegativity, thus directly producing a quinonoid product but not catechol. This compound was identified to be quinone methide tautomer of furanoquinone [63].

Reaction of *o*-quinones with hydroxyl group in phenols (and catechols) may proceed smoothly because of the acidity of the phenolic hydroxyl group (p*K*a ca. 10). Some *o*-quinones undergo 1,4-Michael reaction with the parent catechols to gives rise to dimeric products with a dibenzodioxin skeleton. An example was seen in the tyrosinase-catalyzed oxidation of hydroxytyrosol [(2-(3,4-dihydroxyphenylethanol)) [64] (Figure 14). However, this type of self-coupling reactions may not be a dominant pathway for the fate of *o*-quinones, because the reaction requires high concentrations of the substrate catechol and the product *o*-quinone for an efficient formation of the dimer through a bimolecular reaction. A unique method of preparing the dibenzodioxins was reported by Stratford et al. [65] who witnessed the tyrosinase-catalyzed oxidation of 4-halogenated catechols giving rise to the dibenzodioxin derivatives efficiently.

Reaction of *o*-quinones with hydroxyl group in water or alcohols on the other hand would be difficult to occur (Figure 4). Addition of water molecules to *o*-quinones will produce trihydroxy product that will be rapidly oxidized with the generation of the *p*-quinone product, 2-hydroxy-1,4-benzoquinone (Figure 15). This reaction seems possible for some *o*-quinones. However, in reality, it is difficult to isolate a single product and show the occurrence of such a reaction. With tyrosinase for instance, the nucleophilic addition of water to *o*-quinone products formed during the reaction has not been reported. However, formation of 6-hydroxydopa in peptidyl dopa moiety of topaquinone cofactor contained in amine oxidases appear to proceed by a nucleophilic addition reaction of water to dopaquinone followed by re-oxidation [66]. Whether such a hydroxylation reaction is of enzyme catalyzed or of nonenzymatic origin is yet to be resolved. Hydroxylation of quinones can occur by a different route in some cases. For example, in peroxidase/H_2_O_2_ system hydroperoxide anion can serve as a nucleophile (p*K*a of H_2_O_2_ is 11.6) and cause this reaction. Thus, two group of workers have shown that the peroxidase/H_2_O_2_ oxidation of *N*-acetyldopamine and hydroxytyrosol afforded the 2-hydroxy-1,4-benzoquinone derivatives in good yields [67,68] (Figure 15).

Rhododendrol, 4-(4-hydroxyphenyl)-2-butanol, is a phenol that was widely used as a skin lightening compound until 2013 in Japan. Continuous application of this compound has caused severe leucoderma in many consumers [69]. Although rhododendrol inhibits mushroom tyrosinase [70], it also serves as a good substrate not only for mushroom tyrosinase [71,72] but also for human tyrosinase [73]. In fact, rhododendrol is a better substrate for human tyrosinase than even the natural substrate, L-tyrosine, [73]. There is a possibility that *o*-quinone from rhododendrol might form the six-membered ring cyclic ether (RD-cyclic catechol) through addition of the hydroxyl group to *o*-quinone moiety within the same molecule (Figure 16). However, the feasibility of this intramolecular addition reaction has been questioned by Kishida et al. [74] using density functional theory-based first principles calculations. Nevertheless, the production of rhododendrol-cyclic quinone was unambiguously confirmed because rhododendrol-cyclic catechol was isolated after reduction with NaBH_4_ [71]. Interestingly, rhododendrol-cyclic quinone is gradually converted to rhododendrol-hydroxy-1,4-quinone by the addition of water molecule and decays to oligomeric products. Thus, the mechanism of production of RD-cyclic catechol still remains to be elucidated. 

### 3.5. Redox Exchange of o-Quinones with Reducing Agents

The next important reaction of *o*-quinones that occurs very fast is its redox exchange with reducing agents. These reductions proceed so fast to be of significant importance to biological systems. Ascorbic acid is the most important biological reducing agent because of its high reactivity and abundant levels in various tissues. *o*-Quinones produced in biological systems are often reduced back to catechols by ascorbic acid instantaneously. Such reduction prevents the drastic effects caused by highly reactive quinones. But the disadvantage of this process is ascorbic acid being consumed over and over until it is completely depleted as the reduced catechol can be easily re-oxidized by the endogenous oxidizing enzymes. Thus, oxidative browning of plant products, mushrooms etc., which often accompanies wounding will deplete cellular reduction potential and cause deleterious effects. Again, cellular thiols and other nucleophiles will also undergo addition reaction with quinones thereby adding more damage to the systems. NADH is also able to reduce *o*-quinones to catechols [75]. For synthetic purposes, ascorbate reduction of quinones formed by the tyrosinase-catalyzed oxidation of monophenols proved to be a very convenient and efficient way to synthesize catecholic compounds [75,76]. Some quinones can also be reduced by catechols. In this process, catechol providing the reducing equivalent gets oxidized to form quinone and the quinone substrate is reduced to catecholic product. This process known as redox exchange reaction or double decomposition is biologically very important as it is heavily involved in both eumelanin biosynthesis and pheomelanin biosynthesis. Significance of redox exchange reactions in melanogenesis will be discussed in Section 6.

### 3.6. Reaction of o-Quinone from Hydroquinone

Hydroquinone is a unique phenol in that it belongs to the *para*-diphenols but at the same time to the 4-substituted phenols. In fact, hydroquinone is oxidized to *p*-benzoquinone when oxidized by tyrosinase in the presence of dopa [77]. This reaction proceeds through redox exchange between hydroquinone and dopaquinone. Hydroquinone is cytotoxic to melanocytes and is widely used as a topical hypopigmenting agent [78]. Hydroquinone appears to exert its effects mainly in melanocytes with active tyrosinase activity [79]. Palumbo et al. [77] showed that hydroquinone, as a phenol, acts as a substrate of tyrosinase and generated the hydroxylated product, which was characterized as a red colored chromophore [80] (Figure 17). The red chromophore was identified as 2-hydroxy-1,4-benzoquinone [77]. The immediate oxidation product from hydroquinone by tyrosinase is most likely 4-hydroxy-1,2-benzoquinone, but it is immediately tautomerized to the more stable 2-hydroxy-1,4-benzoquinone [81]. The production of 2-hydroxy-1,4-benzoquinone precludes the generation of catecholic products that may activate tyrosinase (see Section 2), which is why the oxidation of hydroquinone by tyrosinase terminates at an initial phase when a catecholic cofactor is not present [81].

## 4. Reactivity of *o*-Quinones through Quinone Methide Tautomers

### 4.1. Addition of Water to Quinone Methide Tautomers

With 4-alkyl substituted quinones a novel isomerization reaction occurs. The occurrence of this reaction depends on the substituents present on the alkyl group. The simplest 4-alkyl quinone is 4- methyl-*o*-benzoquinone. Isomerization of this compound would generate the *p*-quinone methide, although with this simple compound the chance of isomerization seems to be negligible. However, a number of side chain alkyl substituted compounds can easily isomerize to *p*-quinone methide derivatives. *p*-Quinone methides are tautomers of *o*-quinones with one of the carbonyl oxygen replaced with a benzylic methylene group (Figure 18). The weak acidity of benzylic hydrogen atom renders the conversion to quinone methide tautomers possible. The deprotonation step during the quinone methide formation is proved to be base-catalyzed using deuterium-labeled 4-propionylcatechol [82]. Quinone methides are much more reactive than the corresponding *o*-quinones [6,19,83]. *p*-Quinone methides rapidly add on to any nucleophiles present in the medium. If there are no other nucleophiles present in the medium, they form dimers and other potential oligomers. They are even capable of adding on to water molecules forming side chain hydroxylated products. Both 3,4-dihydroxybenzylalcohol [84] and 3,4-dihydroxybenzylamine [85] are rapidly oxidized to form their corresponding quinone derivatives. These quinones rapidly isomerize to unstable quinone methides which add on to water molecules forming side chain hydroxylated products. The product formed with 3,4-dihydroxybenzylalcohol loses a water molecule in the side chain and produces 3,4-dihydroxybenzaldehyde as the end product [84]. Loss of ammonia from the product of 3,4-dihydroxybenzylamine also produces 3,4-dihydroxybenzaldehyde as final product [85]. 3,4-dihydroxyphenylglycine also exhibits a similar reaction producing 3,4-dihydroxybenzaldehyde as the end product [86]. Another side chain hydroxylation was witnessed during the oxidation of 3,4-dihydroxyphenylacetic acid [61]. The quinone generated form this compound rapidly isomerized and reacted with water producing 3,4-dihydroxymandelic acid, along with other products. Recently, Brooks et al., [87] has demonstrated the addition water to the side chain with 4-ethylphenol. Tyrosinase-catalyzed oxidation of the 4-ethylphenol produced the corresponding quinone which after isomerization and water addition generated 1-(3,4-dihydroxyphenyl)ethanol.

### 4.2. Reaction of Quinone Methides from Catecholamine Metabolites

The significance of quinone methide intermediates in biosynthesis of catecholamines was first proposed by Senoh and Witkop as early as 1959 [88]. They suggested that norepinephrine is produced by the hydration of *p*-quinone methide formed from dopaminequinone. However, this novel concept was subsequently disproved with discovery of dopamine β-hydroxylase, which was shown to catalyze the direct hydroxylation of the β-carbon atom of dopamine by a stereospecific reaction [89]. Much later, Sugumaran and Lipke [90] rediscovered the significance of quinone methides in catecholamine metabolism by demonstrating the possible occurrence of the side chain hydroxylation reaction during cuticular enzyme catalyzed oxidation of a number of catecholic compounds. Although initially they proposed a direct 1,6-oxidation of 4-alkyl catechols to *p*-quinone methides, careful fraction of insect enzyme system revealed that this reaction proceeds with the intermediary formation of 4-alkyl quinone by a tyrosinase-catalyzed oxidation reaction and then the isomerization of 4-alkyl quinones to quinone methides catalyzed by a specific quinone isomerase [91] (Figure 18). Their study revealed that not all 4-alkyl quinone exhibit spontaneous isomerization to *p*-quinone methides. Some such as *N*-acetyldopamine quinone required specific enzyme to catalyze the reaction. Electron withdrawing substituents such as carbonyl and carboxyl groups facilitated the nonenzymatic tautomerization and fully reduced side chains such as -CH_2_-CH_2_-X (X not being an electron withdrawing group) strictly required enzymatic assistance.

Reactivity of *o*-quinones with 4-substituted side chain having hydroxyl or carboxyl group has been extensively studied because they are likely intermediates in sclerotization in insect cuticle [6,19,83] or they are incorporated into neuromelanin in brain [2,92]. Sclerotization is a process in which the cuticle of an arthropod is hardened by cross-linking of the cuticle proteins. 3,4-Dihydroxyphenylethanol (DOPE) and 3,4-dihydroxyphenylethyleneglycol (DOPEG) are alcohol metabolites, while 3,4-dihydroxyphenylacetic acid (DOPAC), and DOMA are carboxylic acid metabolites of catecholamines dopamine and norepinephrine, respectively. Reactivity of DOPE quinone was first reported by Sugumaran et al. [93] using a preparation of cuticular enzyme(s) from *Sarcophaga bullata*. They showed a rapid conversion to DOPEG through the quinone methide intermediate (Figure 19). DOPEG was then gradually converted to 2-oxo-DOPE through another quinone methide intermediate. DOMA quinone, generated either enzymatically [94] or electrochemically [95] was converted to 3,4-dihydroxybenzaldehyde (DHBAld) through the decarboxylative rearrangement giving a quinone methide intermediate. DOPAC quinone also rapidly rearranged to give 3,4-dihydroxybenzylalcohol (DHBAlc) through a quinone methide intermediate, which was then gradually converted to DHBAld through the same quinone methide as from DOMA quinone [84,96]. Ito et al. [97] compared chemical reactivity of *o*-quinones from DOPE, DOPEG, DOPAC, and DOMA to be: DOMA (a half-life, *t*_1/2_, of 3.0 min at pH 6.8) > DOPAC (9.1 min) > DOPE (14.6 min) > DOPEG (30.9 min). It appears that the presence of carboxyl group next to the benzylic carbon in DOPAC and DOMA renders the *o*-quinone intermediates more reactive than in the alcohols DOPE and DOPEG. On the other hand, the presence of hydroxyl groups on the benzylic carbon showed ambiguous effect, either suppressive (DOMA > DOPAC) or accelerating (DOPE > DOPEG). Those *o*-quinones show a common metabolite fate: conversion to quinone methide tautomers followed by secondary reactions such as addition of water molecule or tautomerization to carbonyl group (Figure 19). α-(3,4-Dihydroxyphenyl)lactic acid which is a homolog of DOMA also exhibits the same oxidative decarboxylation reaction and produces 3,4-dihydroxyacetophenone as the end product [98].

In papilionid butterflies, the wing color is produced by the interaction of *N*-β-alanyldopamine with kynurenine. Of the several pigments formed by this interaction, one pigment compound—papilliochrome II, was identified as diastereoisomeric mixtures of kynurenine adduct of *N*- β-alanyldopamine [99,100], the formation of which was biochemically examined by one of our groups [101]. Nonenzymatic addition of the amino group of kynurenine to the *N*-β-alanyldopamine quinone methide enzymatically generated by tyrosinase and quinone isomerase gives rise to diastereoisomers of this compound (Figure 20). During this reaction, water addition to form *N*-β-alanylnorepinephine as a side product is also witnessed [101].

### 4.3. Side Chain Desaturation (Dehydrogenation) Reactions

Another interesting reaction of 4-alkyl quinone is their tendency to undergo side chain desaturation. The side chain desaturation reactions play pivotal role in key biological processes such as melanin biosynthesis and sclerotization of insect cuticle [6,19,83,102,103]. Therefore, we will examine the side chain reactivity of 4-alkyl quinones in this section. Side chain desaturation (dehydrogenation) can proceed when a carbonyl group (ketone or ester) is present at the γ-position of the side chain to make an extension of conjugation system possible through the quinone methide intermediate. The first example of side chain desaturation was witnessed with dihydrocaffeic acid [3-(3,4-dihydroxyphenyl)propionic acid] derivatives (Figure 21). Oxidation of 3,4-dihydrocaffeic acid ester produced its quinone readily, which rapidly isomerized to quinone methide and further exhibited another isomerization to yield side chain desaturated compound, caffeic acid ester [60]. A similar behavior was also exhibited by the methyl amide derivative [104]. Even *N*-acetyldopa esters exhibited this side chain desaturation reactions producing dehydro-*N*-acetyldopa esters [105,106]. Dehydrodopa is a constituent of tunichromes found in blood cells of tunicates, which are implicated in metal sequestering and other functions [107]. The need for another enzyme associated with quinone methide metabolism was discovered when the reactions of *N*-acetyldopamine quinone and isomeric dihydrocaffeiyl methyl amide quinone were compared [104]. While dihydrocaffeiyl methyl amide quinone exhibited rapid isomerization to side chain desaturated product, *N*-acetyldopamine quinone required the assistance of an enzyme. This enzyme was subsequently discovered as *N*-acetyldopamine quinone methide/1,2-dehydro-*N*-acetyldopamine tautomerase [108,109]. Therefore, left unassisted by this isomerase, *N*-acetyldopamine quinone methide simply gives rise to water adduct *N*-acetylnorepinephrine [88,101]. Such a marked difference in the reactivity between *N*-acetyldopa esters and *N*-acetyldopamine can be ascribed to the presence of electron-withdrawing carboxyl group in the former. Dopachrome isomerase reaction represents the second important side chain desaturation reaction. Dopachrome undergoes slow isomerization to its quinone methide and the quinone methide then exhibits side chain desaturation producing 5,6-dihydroxyindole (DHI) and 5.6-dihydroxyindole-2-carboxylic acid (DHICA) [102,103]. More about this reaction will be discussed under melanin biosynthesis (Section 6.2).

Raspberry ketone, 4-(4-hydroxyphenyl)-2-butanone, is a ketone analogue of rhododendrol and known to have caused chemical leukoderma in 1998 [110]. Raspberry ketone quinone undergoes a side chain desaturation giving 3,4-dihydroxybenzalacetone (DBL) quinone due to the presence of a carbonyl group at the γ-position of the side chain [111]. Interestingly, the C2 position of *o*-quinone moiety in DBL quinone becomes more reactive than the C5 position in addition reaction with thiols through the electron-withdrawing effect of 3-oxo-1-butenyl group (Figure 21). A similar activation by a conjugated double bond was observed in resveratrol quinone and chlorogenic acid quinone (Figure 6).

### 4.4. Reactivity of Side Chain Desaturated Quinones

Earlier we examined the side chain desaturation reactions of different catecholamine derivatives. The side chain desaturated compounds play very crucial role in a number of biological process. Three uncyclized unsaturated dopa derivatives that are central to other biological process are—1,2-dehydro-*N*-acyldopamines which are absolutely essential for insect sclerotization [6,83,112,113]; 1,2-dehydro-*N*-acyldopa esters which is partly responsible for gluing properties observed with dopyl proteins [114], and 1,2-dehydrodopa which forms a basic unit in a number of tunichrome derivatives that could act as antibiotic compounds and cementing material [5,62,107,115]. 1,2-Dehydro-*N*-aceyldopamines upon tyrosinase oxidation may generate the quinone as the immediate product but characterizing the quinone even transiently was practically impossible due to the fact that it instantaneously isomerized to tautomeric quinone methide imine amide [19,112,113,116,117] (Figure 22). These compounds exhibited rapid reaction with the parent catechol generating benzodioxan dimers and other oligomers. 1,2-Dehydro-*N*-acyldopa was even more reactive in generating the quinone methide product [62]. As stated earlier, oxidation of 1,2-dehydro-*N*-acetyldopa resulted in rapid quinone formation followed by intramolecular cyclization. The dihydroxy coumarin thus formed further suffered more oxidation in the reaction mixture and generated a number of polymeric derivatives *via* another quinone methide imine amide derivative. Surprisingly, 1,2-dehydro-*N*-acyldopa ester produced the corresponding quinone and not the quinone methide tautomer as the immediate product [114]. Irrespective of the nature of the quinonoid species, all of these compounds exhibited oxidative polymerization to form different benzodioxan derivatives. These transformations attest the versatile reactivity of side chain substituted quinones (see [19] for more details).

### 4.5. Reaction of Estradiol Quinones

Metabolic fate of 17-β-estradiol, a phenolic steroid hormone, was examined by Pezzella et al. [118] showing the involvement of quinone methide intermediates. Tyrosinase-catalyzed oxidation of 17-β-estradiol afforded catechol estrogens, 2-hydroxyestradiol and 4-hydroxyestradiol at the low, but physiologically relevant substrate concentrations of 1–10 nM [118]. In addition, they isolated metabolites, 6-oxo-2-hydroxyestradiol, 9,11-dehydro-2-hydroxyestradiol, 6,7-dehydro-2-hydroxyestradiol, and 9,11-dehydro-4-hydroxyestradiol (Figure 23). At higher estradiol concentrations, e.g., 1 µM, additional products including 6,7,8,9-dehydro-2-hydroxyestradiol and dimers were also detected. These products arise from quinone methide intermediates. Reactivity of quinone (or quinone methide) from estradiol with DNA is discussed in the following section.

## 5. Reactivity of *o*-Quinones with Macromolecules

### 5.1. Covalent Binding with Proteins

The high reactivity of thiol group renders proteins to react with *o*-quinones. This type of protein modifications is biologically important because this may lead to denaturation of thiol proteins and inhibition of thiol enzymes, leading to cytotoxicity [48,53]. Kato et al. [119] found that dopaquinone reacts with thiol group in cysteine residue of thiol proteins, bovine serum albumin, alcohol dehydrogenase, and isocitrate dehydrogenase in yields of 5.4, 44, and 33%, respectively. The yields can be determined by acid hydrolysis to release cysteinyldopas (or other cysteinylcatechols; Figure 24). This hydrolysis method is generally applicable to evaluate binding of thiol proteins with *o*-quinones, e.g., rhododendrol quinone [120] and *o*-quinone from *N*-propionyl-4SCAP [121] (Figure 24). During acid hydrolysis, some functional groups undergo additional modifications such as the substitution of a chlorine group in RD and the hydrolytic cleavage of an amide group in *N*-propionyl-4SCAP (Figure 24).

Reactivity of biologically relevant *o*-quinones with bovine serum albumin was compared by Ito et al. [123] showing the reactivity of *o*-quinones from catecholamines to be dopamine > norepinephrine > dopa > epinephrine; this reactivity is inversely correlated to the rate of cyclization of the side chain amino group (see above).

The covalent binding of *o*-quinones with thiol proteins has been proposed as the mechanism of cytotoxic action of various 4-substituted phenols in melanocytes and production of neo-antigens. It should be stressed that the initial step in this cytotoxic and immunologic action of phenols is their conversion to *o*-quinones through tyrosinase-catalyzed oxidation [124]. The covalent binding of rhododendrol quinone to proteins through cysteine residues was found to be 20- to 30-fold greater than dopaquinone in B16F1 mouse melanoma cells [120] (Figure 24). The addition of thiol proteins to rhododendrol quinone was also confirmed in vivo in a mouse model mimicking Japanese skin [122]. Monobenzone (hydroquinone benzyl ether) is a 4-substitued phenol known for inducing whitening of the skin due to loss of melanocytes. It serves as a good substrate for tyrosinase [124,125] and the *o*-quinone produced reacts not only with small thiols but also the sulfhydryl group in bovine serum albumin [125]. It was proposed that monobenzone quinone, acting as a hapten, reacts with thiol proteins to form a neo-antigen that triggers immune response leading to a specific loss of melanocytes [125,126,127] One possible mechanism of rhododendrol-induced leukoderma is the binding of rhododendrol quinone with thiol proteins that leads to immune response through antigen presentation or ER stress through inactivation/denaturation of thiol enzymes/proteins [72]. 4SCAP, *N*-propionyl-4SCAP, and related phenolic thio-ethers were examined for inhibiting the growth of malignant melanoma in vitro and in vivo, in the hope for developing antimelanona agents [53,121,128].

Li et al. [54] examined reactivity of 4-methyl-*o*-benzoquinone with bovine serum albumin, human serum albumin, and α-lactalbumin (no cysteine residue) showing their second order rate constants of 3.1 × 10^4^ M^−1^ s^−1^, 4.8 × 10^3^ M^−1^ s^−1^, and 4.0 × 10^2^ M^−1^ s^−1^, respectively. When we consider the large excess of amine/guanidine groups in proteins (83 vs. one cysteine residue in serum albumin), it is the cysteine residues that are kinetically the most important targets in proteins. The reactivity of dopaminequinone with proteins was recently re-visited by Wakamatsu et al. [129] showing that the efficacy of binding of proteins through cysteine residue depends on its reactivity (or steric availability), e.g., 70% in bovine serum albumin and 20% in β-lactoglobulin, both containing one cysteine residue.

3,4-Dihydroxybenzaldehyde (DOPAL) is formed from enzymatic oxidation of dopamine by monoamine oxidase. DOPAL is toxic to cells and animals such that DOPAL accumulation might contribute to neuronal malfunctions and eventual loss of dopaminergic neuron [130]. DOPAL itself reacts with *N*-*α*-acetyllysine to form the reversible Schiff base but not with *N*-acetylcysteine [131] (Figure 25). Jinsmaa et al. [132] compared reactivity of dopaminequinone and DOPAL quinone showing that DOPAL quinone is more efficient than dopaminequinone in modifications of α-synuclein, a protein whose oligomerization plays a role in pathogenesis of Parkinson disease (Figure 25). DOPAL quinone appears highly reactive due to the presence of both *o*-quinone and aldehyde groups [131], but its reactivity toward various nucleophiles remains to be studied. 

### 5.2. Covalent Binding with DNA

DNA is also susceptible to modification by *o*-quinones. Cavalieri et al. [58] showed that dopaminequinone, formed with tyrosinase, horseradish peroxidase, or liver microsomes, reacted with DNA to form the depurinating dopamine adducts, dopamine-6-N3Ade and dopamine-6-N7Gua, albeit at low yields (Figure 12). This type of DNA modification would lead to mutation of DNA initiating cancer and other diseases such as Parkinson’s disease in the case of dopaminequinone [133]. *o*-Quinones from estradiol, estradiol-3,4-quinone and estradiol-2,3-quinone, were shown to react with DNA to form the depurinating adducts [134]. Estradiol-3,4-quinone afforded the 1-N3Ade and 1-N7Gua adducts, whereas estradiol-2,3-quinone afforded the 6-N3Ade adduct (Figure 26). Estradiol-3,4-quinone was found to be much more reactive than estradiol-2,3-quinone. Interestingly, the former reacted through the *o*-quinone while the latter through the quinone methide tautomer. It is possible that this DNA modification by estradiol quinones might lead to breast, prostate, and other human cancers [134].

## 6. Melanogenesis in Relation to *o*-Quinone Chemistry

### 6.1. Early Stages of Mixed Melanogenesis

Melanogenesis is a complex pathway leading to the production of the dark brown to black eumelanin and the yellow to reddish-brown pheomelanin [1,135,136,137,138,139,140,141]. The early stages of eumelanin production starting from tyrosine were established by the pioneer works of Raper [142] and Mason [143], and are thus called “Raper-Mason pathway”, while those of pheomelanin production were elucidated by Prota, Nicolaus, and collaborators in the late 1960s [135,136,137,144]. However, most of natural melanin pigments are actually co-polymers or mixtures of eumelanin and pheomelanin in varying ratios [145] and thus melanin production can be considered as “mixed melanogenesis” [1,146]. Therefore, a unified scheme for mixed melanogenesis was proposed to account for how this complex process is chemically regulated. 

The early stages of mixed melanogenesis are now well elucidated using the technique called pulse radiolysis [17,147]. Pulse radiolysis is a powerful tool to study the chemical fate of highly reactive melanin precursors. Pulse radiolysis of N_2_O-saturated KBr solution produces dibromine radical anions Br_2_^−^ within 1 nanosecond. Addition of dopa to this system produces dopasemiquinone which then disproportionates to generate dopaquinone and dopa within around a millisecond (Figure 27). Then the fate of dopaquinone can be followed by spectrophotometry in the presence or absence of a target molecule such as cysteine.

When cysteine is not present, the first step (r1 = 3.8 s^−1^) in eumelanogenesis proceeds through the intramolecular, 1,4-Michael addition of the amino group to produce cyclodopa (leucodopachrome) [148]. This cyclization is a base-catalyzed reaction as shown by the rate constants of 0.91 s^−1^ at pH 6.6 and 7.6 s^−1^ at pH 7.6, because the amino group needs to be a non-protonated –NH_2_ [149]. As cyclodopa is formed, it is rapidly oxidized by dopaquinone to form dopachrome through a redox exchange giving dopa (r2 = 5.3 × 10^6^ M^−1^ s^−1^) [150]. When cysteine is present, the first step in pheomelanogenesis proceeds very quickly through the intermolecular, 1,6-Michael addition of cysteine (r3 = 3 × 10^7^ M^−1^ s^−1^) [149]. The second step in pheomelanogenesis is the redox exchange with dopaquinone to generate cysteinyldopa quinone, which proceeds at a slower rate (r4 = 8.8 × 10^5^ M^−1^ s^−1^) [148]. It should be emphasized that the chemical reactivity of dopaquinone controls these four reactions through an intramolecular addition of amino group, an intermolecular addition of sulfhydryl group, and two redox exchange reactions. From these kinetic data, an “Index of Divergence” (D) between eumelanogenesis and pheomelanogenesis can now be derived. By taking dopachrome and cysteinyldopa quinone as representatives of those two pathways, Land et al. [150] proposed the formula: D = r3 × r4 [cysteine]/r1 × r2.

This leads to a “crossover value” (i.e., for D = 1) for switching between eumelanogenesis to pheomelanogenesis when cysteine concentration at the site of melanogenesis reaches 0.8 µM. 

### 6.2. Late Stages of Mixed Melanogenesis

Dopachrome accumulated during early stages of eumelanogenesis. The orange-red pigment dopachrome is fairly stable having a half-life of about 30 min (first-order rate constant of 4.0 × 10^−4^ s^−1^). It spontaneously decomposes to give mostly 5,6-dihydroxyindole (DHI) by decarboxylation at neutral pH values (Figure 28). The ratio of DHI to DHICA produced in the absence of dopachrome tautomerase (DCT) or metal cations is 70:1 [151]. However, in the presence of DCT (Tyrp2), dopachrome undergoes tautomerization to preferentially produce DHICA [152,153]. Chemically, dopachrome solution prepared by ferricyanide oxidation of DL-dopa afforded DHI at pH 6.5 and DHICA at pH 13, which is a basis of preparation of those dihydroxyindoles at sub-gram scales in ca. 70% yields [138,154].

Ito et al. [155] examined in detail the effects of pH and Cu^2+^ ions on the conversion of dopachrome to DHI and DHICA and their subsequent oxidation to form eumelanin. The results indicate that acidic pHs suppress these late stages of eumelanogenesis and Cu^2+^ ions greatly accelerate the conversion of dopachrome to DHICA (and hence increase the ratio of DHICA to DHI) and its subsequent oxidation. The conversion of dopachrome to DHI or DHICA was first suggested to involve the quinone methide intermediate by Sugumaran’s group [103,141]. Sugumaran and Semensi [102] examined the enzymatic conversion of dopachrome, dopachrome methyl ester, and α-methyldopachrome methyl ester. In the presence of a dopachrome-converting factor isolated from the hemolymph of insect, *Manduca sexta*, dopachrome was converted to DHI while dopachrome methyl ester was tautomerized to DHICA methyl ester. At a weakly alkaline pH of 8.0 without any enzyme, a-methyldopachrome methyl ester produced a stable quinone methide tautomer [100]. However, at the physiological pH value, the enzyme catalyzed this reaction [102]. These results conclusively proved the transient production of quinone methide tautomer during conversion of dopachrome to DHI (Figure 28). Formation of the same quinone methide intermediate during the conversion of dopachrome to DHICA is similarly likely, but it has not yet been demonstrated.

Then, how can DHI and DHICA be oxidized finally to give rise to eumelanin? The redox exchange between DHI and dopaquinone was found to proceed but not quite to completion with a rate constant of r5 = 1.4 × 10^6^ M^−1^ s^−1^ [156] (Figure 29). This is in the same rate as the rate constants with cyclodopa (r2) and with 5SCD (r4). The reaction with DHICA is much slower (r6 = 1.6 × 10^5^ M^−1^ s^−1^) and does not go to completion. It thus appears that during eumelanogenesis, DHI oxidation proceeds by redox exchange with dopaquinone but DHICA oxidation may require its oxidation to the quinone form catalyzed by tyrosinase [157] or tyrosinase-related protein 1 [158,159]. DHI quinone and DHICA quinone can readily dimerize with the parent DHI and/or DHICA to give homodimers and heterodimers (for a heterodimer, see [160]). Various oligomers up to the stage of tetramers have been isolated and characterized in the form of acetylated derivatives for DHI or non-derivatized form for DHICA [161,162,163].

The late stages of pheomelanogenesis beyond cysteinyldopa (CD) quinones are summarized in Figure 30. Conversion of cysteinyldopa to its dehydrated form, the dihydro-1,4-benzothiazine-3-carboxylic acid (DHBTCA), takes place under biomimetic conditions [164,165]. Interestingly, during tyrosinase oxidation of dopa in the presence of equimolar cysteine, cysteinyldopa isomers are the first major products that are then dehydrated to give DHBTCA isomers [166]. DHBTCA (isomers) accumulates transiently and then decay rapidly in the course of pheomelanogenesis. 

Then how is DHBTCA produced and then decayed? Kinetic experiments using the pulse radiolysis technique [148,149,167] showed that: (1) cysteinyldopa accumulates in the early stage of pheomelanogenesis because the addition reaction of cysteine to dopaquinone producing cysteinyldopa (r3) proceeds much faster than the redox exchange of cysteinyldopa with dopaquinone giving cysteinyldopa quinone (r4), (2) cysteinyldopa is gradually oxidized to cysteinyldopa quinone through the redox exchange with dopaquinone only after cysteine concentration becomes low, (3) the intramolecular cyclization of cysteinyldopa quinone produces the *o*-quinone imine through Schiff base formation (r7 = 10 s^−1^) [168], and (4) the quinone imine then rapidly rearranges with/without decarboxylation to afford the 1,4-benzothiazine (BT) and the 1,4-benzothiazine-3-carboxylic acid (BTCA)(r8 = 6.0 s^−1^) [168]. When the rate of production of cysteinyldopa quinone (and hence the quinone imine) is low as in the case of tyrosinase-catalyzed oxidation, the quinone imine is reduced to DHBTCA by the excess cysteinyldopa, which in turn is oxidized back to cysteinyldopa quinone. It should be stressed that the redox exchange of dopaquinone with DHBTCA to produce the quinone imine (and dopa) is an additional role of dopaquinone [166]. This reaction acts as a driving force in this stage of pheomelanogenesis. 

The last stage of pheomelanogenesis is the oxidative polymerization of BT and BTCA, which eventually leads to the production of pheomelanin (Figure 30). This process appears to be very complex [169] but involves secondary modifications of the benzothiazine moiety leading to the formation of the 3-oxo-3,4-dihydro-1,4-benzothiazine (ODHBT) and the benzothiazole (BZ) [170]. These intermediates, BT (and BTCA), ODHBT, and BZ, are gradually incorporated into the pheomelanin polymer [166]. Using chemical degradative approach [166,171,172], we were able to show that the benzothiazine moiety gradually degrades to form a benzothiazole moiety in the last stage of pheomelanogenesis. This is consistent with the observation by Napolitano et al. [170] that the chromophore of pheomelanin (λ_max_ around 305 nm) is better characterized by a prevalent presence of the benzothiazole moiety (λ_max_ at 303 nm) than the dihydrobenzothiazine (BT and BTCA) moiety (λ_max_ above 340 nm). Recent studies have shown that melanogenesis is suppressed at acidic pH in melanosomes, the organelles specialized in melanin synthesis in melanocytes [173,174]. The catalytic activities of human tyrosinase are progressively suppressed by lowering the pH [174], with a shift to a more pheomelanic phenotype [173]. Recently, we compared the effect of pH on mixed melanogenesis in tyrosinase-catalyzed oxidation of dopa or tyrosine in the absence or presence of cysteine [175]. The results showed that pheomelanin production was greatest at pH values of 5.8 to 6.3, while eumelanin production was suppressed at pH 5.8 compared to pH 6.3–7.3. This suggests that mixed melanogenesis is chemically shifted to more pheomelanic states at weakly acidic pH. This fits well with the kinetic data for the cyclization of 5SCD quinone to the quinone imine (QI), which is promoted by acid (r7 = 15 s^−1^ at pH 5.6 and 3.3 s^−1^ at pH 7.7; [149]).

### 6.3. Metabolic Fates of Catecholamine Quinones

Neuromelanin is a brown to black-brown insoluble melanin-like pigment present in the midbrain of humans and other primates and is known to accumulate during aging [2,21,176]. Dopamine and norepinephrine are considered to be the major precursors of neuromelanin present in the *substantia nigra* and *locus coeruleus* of the brain, respectively [2,176,177]. However, biosynthesis of these neuromelanin appears a complex process and out of the scope of this review. Here, we summarize briefly what are known about the metabolic fates of dopaminequinone and norepinephrinequinone. 

It has been a matter of debates whether tyrosinase plays a significant role in the production of neuromelanin [178]. However, it is very likely that the first step of the neuromelanin production is the oxidation of catecholamines to *o*-quinone derivatives through interaction with redox active metal ions such as Fe^3+^ and Cu^2+^ [2,3,129,176]. Reactivity of dopaminequinone with small thiols and thiol-proteins has been extensively studied [3,129]. Because dopaminequinone cyclizes more slowly than dopaquinone, it binds to proteins through the cysteinyl residues more efficiently [123,129]. On the other hand, in the absence of thiols, dopaminequinone cyclizes and is oxidized to form dopaminechrome in nearly quantitative yield [129,179] (Figure 31). Jimenez et al. [179] showed that tyrosinase oxidation of dopamine proceeds in a similar manner to that of dopa, i.e., cyclization of dopaminequinone followed by redox exchange of the resulting cyclodopamine with dopaminequinone to form dopaminechrome. Dopaminechrome is then slowly polymerized to give rise to dopamine melanin *via* DHI [67]. In addition to the cyclization pathway, in the presence of H_2_O_2_, tyrosinase-catalyzed oxidation of dopamine gave rise to 6-hydroxydopamine in several percent yield [67]. In a model experiment, peroxidase/H_2_O_2_ oxidation of *N*-acetyldopamine gave up to 55% yield of the 2-hydroxy-1,4-benzoquinone derivative (Figure 31). The slow rate of cyclization of dopaminequinone as compared to norepinephrinequinone has been shown by spectrophotometry [179,180] and by cyclic voltammetry [47]. This may possibly lead to the generation of the more reactive quinone methide tautomer of dopamine, which should give rise to secondary products including norepinephrine through the addition of water molecule [67]. However, this aspect of dopaminequinone chemistry has not been examined in detail. In this relation, when dopamine is oxidized by tyrosinase, the yield of dopamine melanin is not high, less than 50% [33], whereas the yield of dopa melanin is nearly quantitative [138]. It is possible that dopaminequinone and/or dopaminechrome may give rise to various secondary, soluble metabolites, similar to those from NE (see below). Certainly, more studies appear necessary to clarify these aspects of dopamine oxidation. In this regard, it should be mentioned that in recent years polydopamine formed by oxidation of dopamine at pH 8.5 has drawn great interests from material scientists because of its unique properties to adhere to all type of surfaces [181,182]. However, structural basis of polydopamine has not been fully clarified as it is so for dopamine melanin [183,184].

The reactivity of norepinephrinequinone remains less studied than that of dopaquinone and dopaminequinone. Jimenez et al. [180] showed that tyrosinase oxidation of norepinephrine proceeds in a similar manner to that of dopa and dopamine but with a faster rate of cyclization. Terland et al. [185] compared reactivity of one-electron oxidation products of catecholamines produced by ferricyanide oxidation. The rate of formation of the corresponding aminochromes was found to be dopamine < norepinephrine < epinephrine. Manini et al. [186] examined the fate of *o*-quinone produced by Fenton reagent (Fe^2+^-EDTA/H_2_O_2_) and horseradish peroxidase/H_2_O_2_. The cyclization pathway proceeds, giving norepinephrinechrome and ultimately norepinephrine melanin. In addition, they were able to detect 3,4-dihydroxymandelic acid, 3,4-dihydroxybenzaldehyde, 3,4-dihydroxybenzoic acid, and 3,4-dihydroxyphenylglyoxylic acid in about 50% combined yields (Figure 32). It would be interesting to examine the metabolic fate of norepinephrinequinone produced by tyrosinase to exclude the effect of H_2_O_2_.

Oxidation of epinephrine (adrenaline) affords a red pigment, called adrenochrome (Figure 33). There were considerable interests in the detection of this aminochrome in various tissues in the 1960s [187]. Adrenochrome seems to be the most stable among various aminochromes at neutral pH [188,189] but at alkaline pH it gradually rearranges to give a highly fluorescent adrenolutin (3,5,6-trihydroxy-*N*-methylindole) [190]. The subsequent metabolic fate remains little known. Because of its stability, adrenochrome can be quantified in biological samples by various methods including HPLC-photodiode array [189].

## 7. Conclusions

Tyrosinase is able to catalyze oxidation of phenols and catechols (*o*-diphenols) to *o*-quinones. In this review, we summarized chemical fates of *o*-quinones. Some of the reactions of *o*-quinones proceed extremely fast with a half-life of less than one second. Some reactions proceed only after tautomerization to the quinone methide form. Because of their high reactivity, *o*-quinones are able to modify the structure of proteins and DNA, leading possibly to cytotoxicity and carcinogenesis. Dopaquinone plays pivotal roles in melanogenesis; it acts chemically in several key steps including the redox exchange with cyclodopa to form dopachrome in the early stages of eumelanin production and the addition of cysteine to initiate pheomelanin production.

During the preparation of this review, the authors have noticed some discrepancies in metabolic fates of *o*-quinones from one study to another. Although most of those discrepancies may arise from differences in the substrate concentration and the pH of the medium, more studies certainly remain to be performed with rigorous identification of the products.

## Figures and Tables

**Figure 1 ijms-21-06080-f001:**
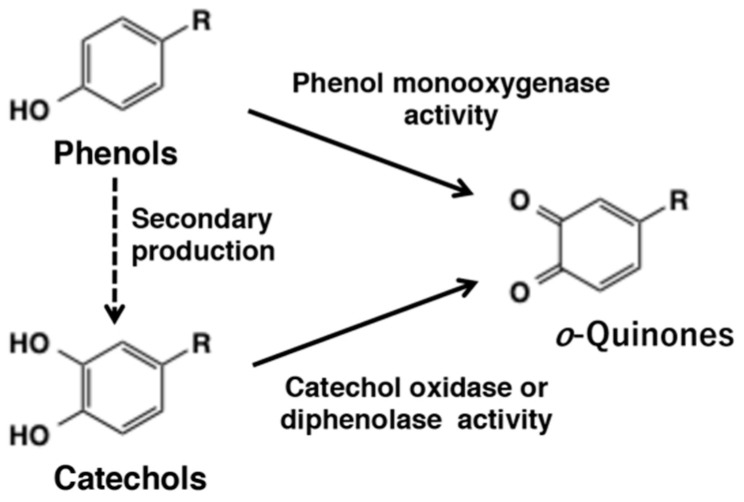
Tyrosinase-catalyzed oxidation of 4-substituted phenols and the corresponding catechols producing *o*-quinones. Note that catechols are not produced directly from phenols [7].

**Figure 2 ijms-21-06080-f002:**
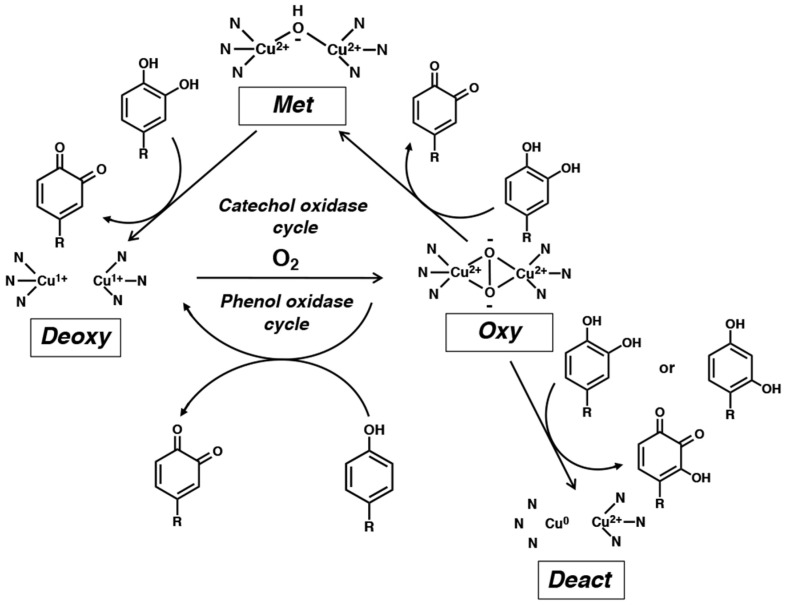
Relationship among the four oxidative states of tyrosinase [7].

**Figure 3 ijms-21-06080-f003:**
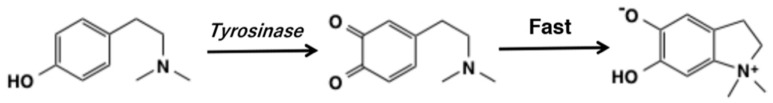
Tyrosinase-catalyzed oxidation of *N*,*N*-dimethyltyramine [11]. This reaction requires a catalytic amount of L-dopa. The *o*-quinone rapidly gives the indolium salt in a spontaneous reaction. This salt does not have an activity to activate *met*-tyrosinase and, thus, the oxidation stops at this stage.

**Figure 4 ijms-21-06080-f004:**
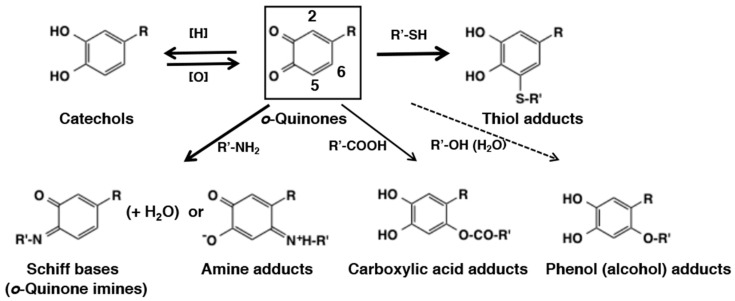
Summary of intrinsic chemical reactivity of *o*-quinones. The relative reactivity is shown in the thickness of the arrows. Note that the reactions with carboxylic acids and alcohol (water) are possible only when the functional groups are present in the side chain of *o*-quinones. For the sake of simplicity, the numbering on the *o*-quinone ring is made so that R- is attached to the C1 position. The addition of thiols proceeds mostly at the C5 (major) and C2 (minor) positions (1,6-Michael addition) while the addition of other nucleophiles proceeds mostly at the C6 position (1,4-Michael addition).

**Figure 5 ijms-21-06080-f005:**
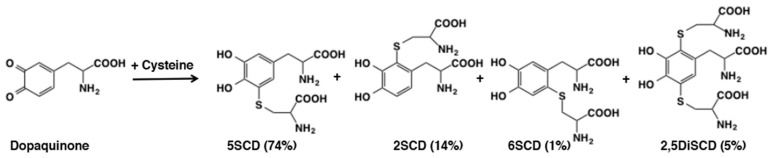
Reaction of dopaquinone with cysteine. 5-*S*-Cysteinyldopa (5SCD) is the major isomer [26] and serves as a biochemical marker of malignant melanoma [28].

**Figure 6 ijms-21-06080-f006:**
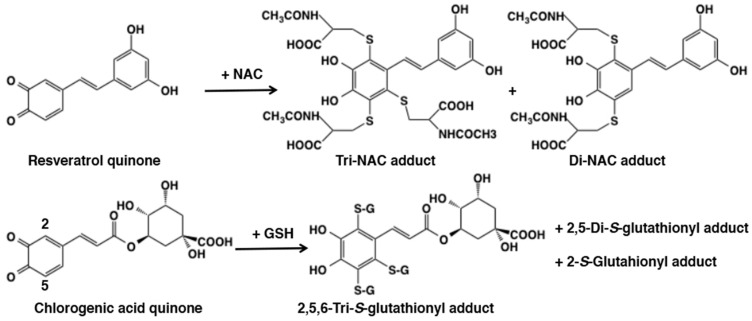
Reaction of resveratrol quinone and chlorogenic acid quinone with thiols [35,37]. Note that the C2 position is most reactive.

**Figure 7 ijms-21-06080-f007:**
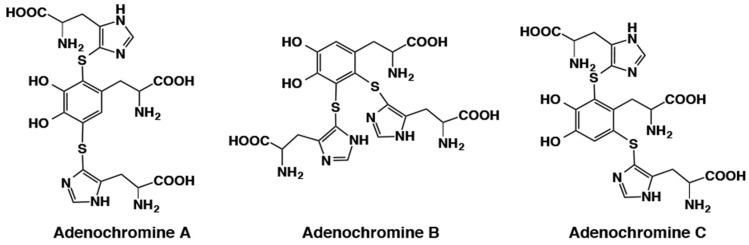
Three structural isomers of di-adducts of 5-thiohistidine to dopa isolated from *Octopus vulgaris* [39,40]. Note that adenochromine A, B, and C correspond to the 2,5-, 5,6-, and 2,6-adducts, respectively.

**Figure 8 ijms-21-06080-f008:**
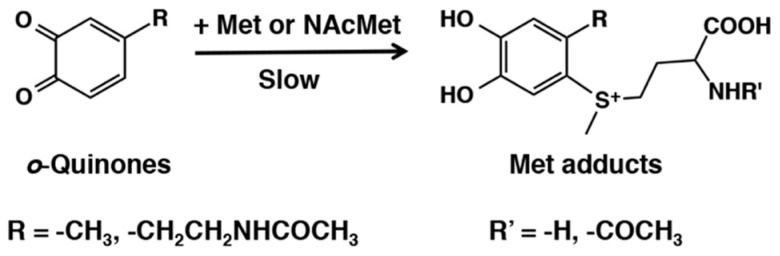
Reaction of *o*-quinones with methionine and *N*-acetylmethionine [46]. Note that the position of attachment of thio-ether group has not been confirmed.

**Figure 9 ijms-21-06080-f009:**
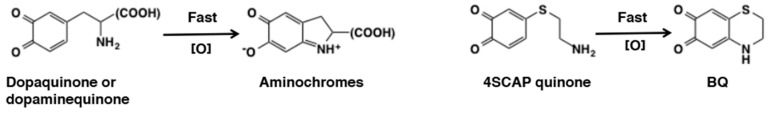
Intramolecular addition of amino group to *o*-quinone group to form aminochromes. A unique example of the intramolecular addition of amino group to form a six-membered ring is presented for 4SCAP quinone [50].

**Figure 10 ijms-21-06080-f010:**
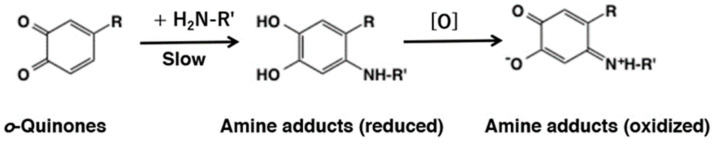
Reaction of *o*-quinones with amines. Note that the oxidized form of amine adducts are more stable than the reduced form [54].

**Figure 11 ijms-21-06080-f011:**
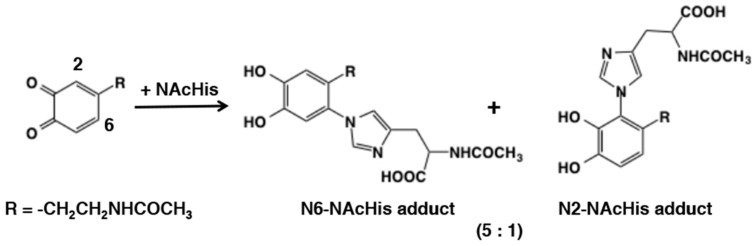
Reaction of *o*-quinones with *N*-acetylhistidine. Note that the position of attachment of the imidazole group was confirmed [57].

**Figure 12 ijms-21-06080-f012:**
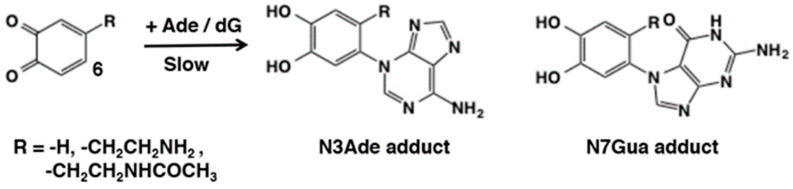
Reaction of *o*-quinones with DNA bases [58]. These adducts are isolated in the reduced form.

**Figure 13 ijms-21-06080-f013:**
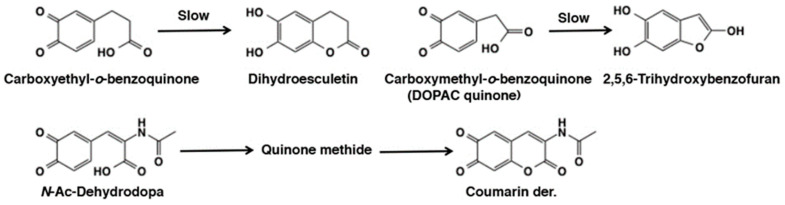
Intramolecular addition of carboxylate group to *o*-quinone group to form lactones [60,61,62].

**Figure 14 ijms-21-06080-f014:**
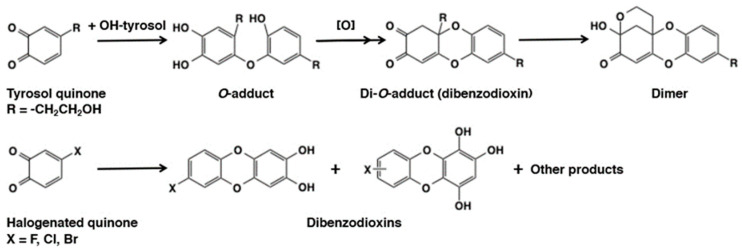
Self-coupling of tyrosol quinone with hydroxytyrosol (HT) to form a dibenzodioxin [64]. An isomer differing in the position of the substituent (R-) was also isolated. An efficient formation of dibenzodioxins was reported for 4-halogenated quinones [65].

**Figure 15 ijms-21-06080-f015:**
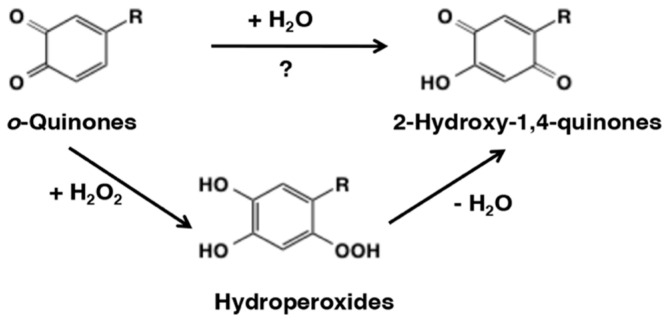
Reaction of *o*-quinones with hydrogen peroxide leading the production of 2-hydroxy-1,4-quinones [67,68]. Note that the direct addition of water molecule to *o*-quinone is unlikely to occur.

**Figure 16 ijms-21-06080-f016:**
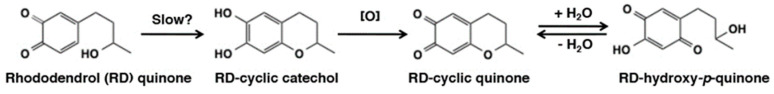
Reaction of rhododendrol (RD) quinone to form RD-cyclic catechol [71]. Note that this cyclization reaction is less likely to proceed [72,74].

**Figure 17 ijms-21-06080-f017:**
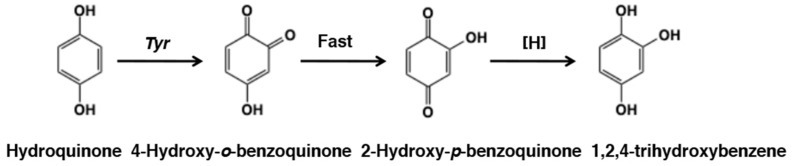
Reaction of hydroquinone in the tyrosinase-catalyzed oxidation [77,80,81].

**Figure 18 ijms-21-06080-f018:**
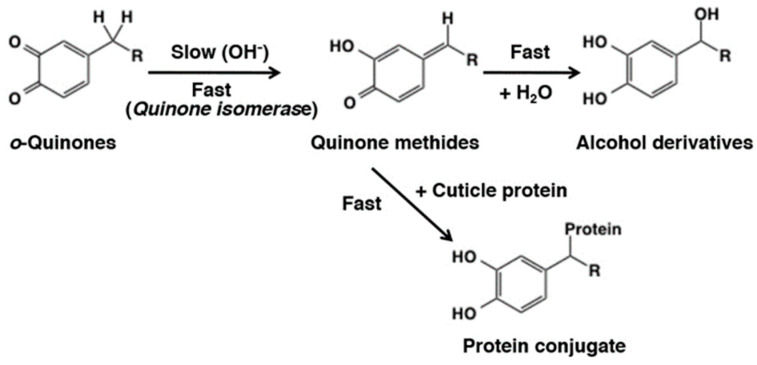
Quinone methide formation from *o*-quinones and the subsequent addition of water molecule to form alcohol derivatives [84].

**Figure 19 ijms-21-06080-f019:**
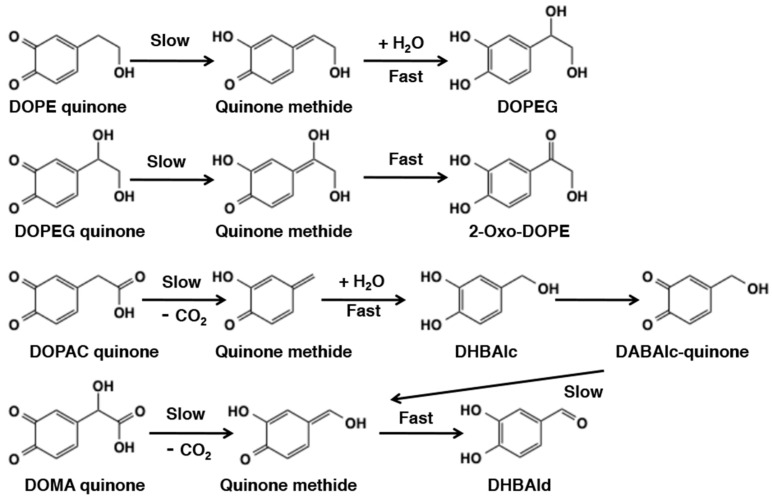
Reaction of *o*-quinone products from catecholamine metabolites 3,4-Dihydroxyphenylethanol (DOPE), 3,4-dihydroxyphenylethyleneglycol (DOPEG), 3,4-dihydroxyphenylacetic acid (DOPAC), and 3,4-dihydroxymandelic acid (DOMA) [94]. Tyrosinase-catalyzed oxidation of the catechols was terminated by the addition of ascorbic acid. Note that the formation of quinone methides proceeds slowly, which is quickly followed by the addition of water molecule or the formation of carbonyl group.

**Figure 20 ijms-21-06080-f020:**
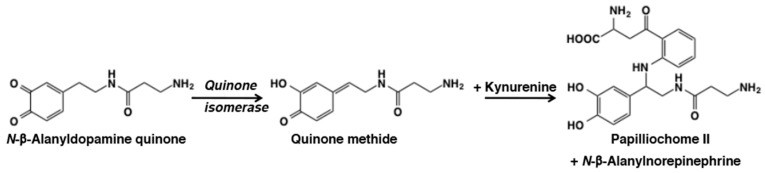
Reaction of *N*-β-alanyldopamine quinone with kynurenine to form papilliochrome II [99,100,101]. Note that quinone isomerase was required for the production of papilliochrome II and that *N*-β-alanylnorepinephrine was also produced through addition of water.

**Figure 21 ijms-21-06080-f021:**
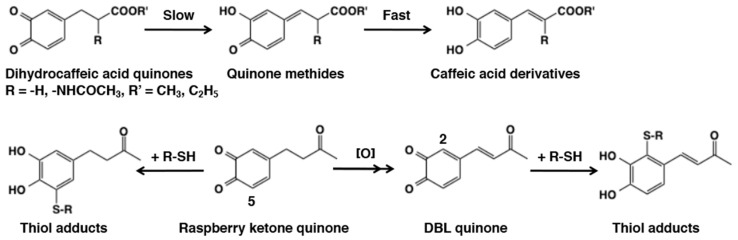
Side chain desaturation from quinone methides from dihydrocaffeic acid quinones [60] and raspberry ketone quinone [111]. Note that dihydrocaffeic acid (caffeic acid) with the *N*-acetyl group can be considered as dopa (dehydrodopa) derivative. The position of thiol addition differs between raspberry ketone quinone and DBL quinone.

**Figure 22 ijms-21-06080-f022:**
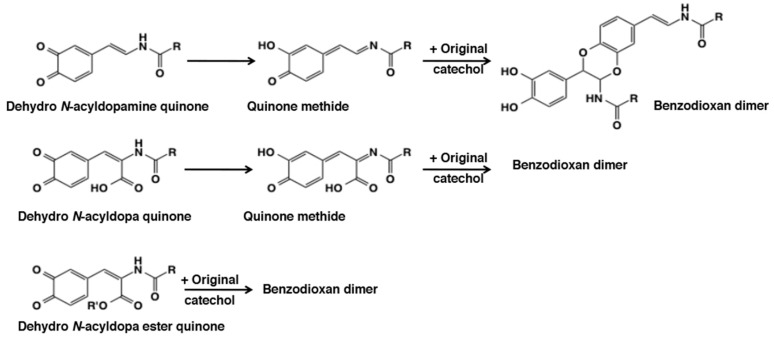
Reaction of side chain desaturated quinones producing benzodioxan dimers [19,62,113,114,115,116,117].

**Figure 23 ijms-21-06080-f023:**
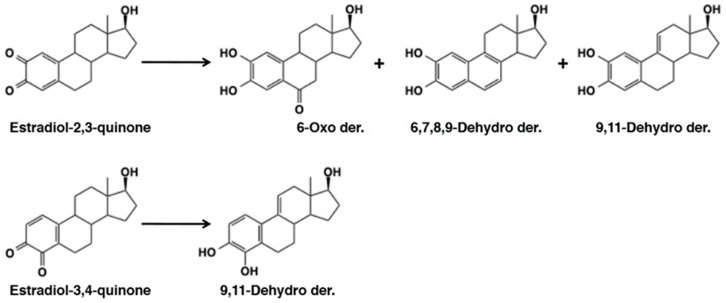
Reaction of β-estradiol quinones [118]. Note that those quinones undergo tautomerization to quinone methides which give rise to the secondary products.

**Figure 24 ijms-21-06080-f024:**
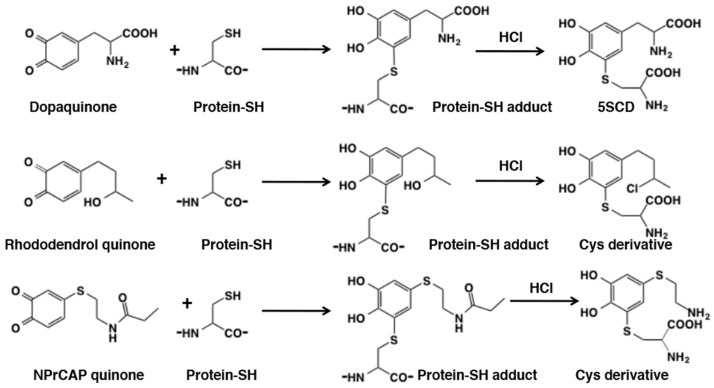
Reaction of *o*-quinones with protein-SH and acid hydrolysis of the adducts to liberate cysteinyl derivatives [30,120,121,122].

**Figure 25 ijms-21-06080-f025:**
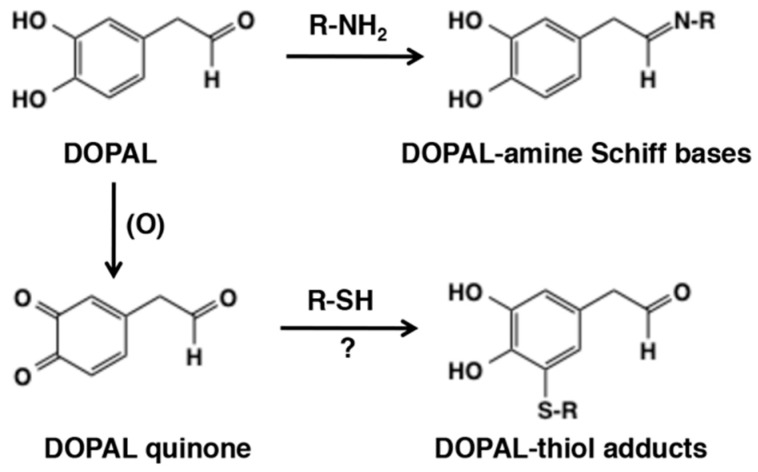
Reaction of 3,4-Dihydroxybenzaldehyde (DOPAL) and its quinone with amines and thiols [131]. Note that the reaction with thiols requires oxidation to the quinone form.

**Figure 26 ijms-21-06080-f026:**
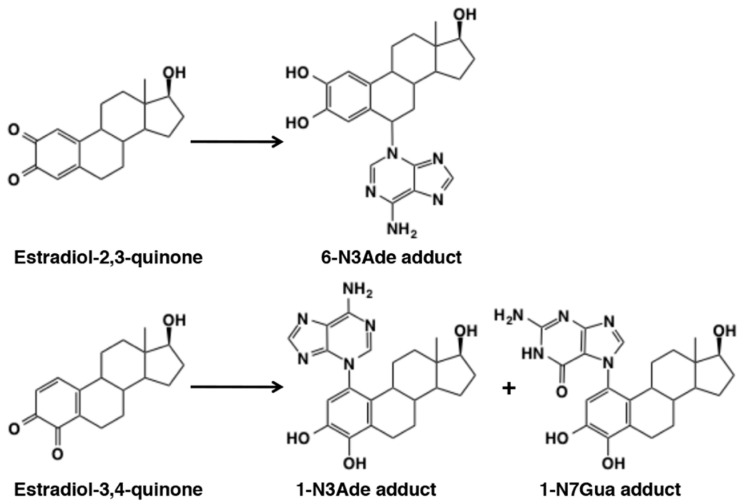
Reaction of β-estradiol quinones with DNA to produce depurinating products [134].

**Figure 27 ijms-21-06080-f027:**
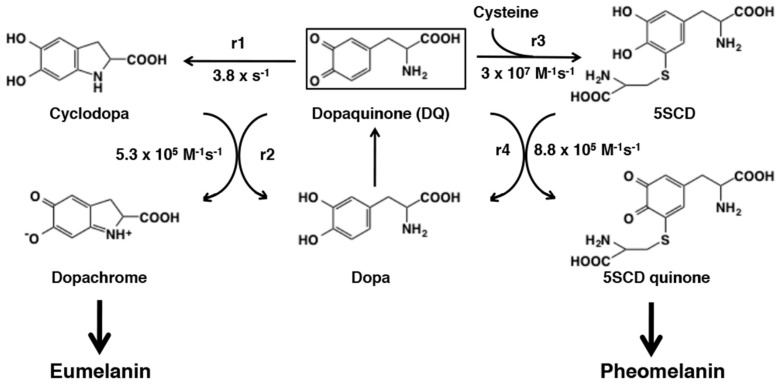
Kinetics of early stages of mixed melanogenesis. Note that the rate constants r1–r4 are controlled by the intrinsic chemical reactivity of dopaquinone (DQ) [17,148,149,150].

**Figure 28 ijms-21-06080-f028:**
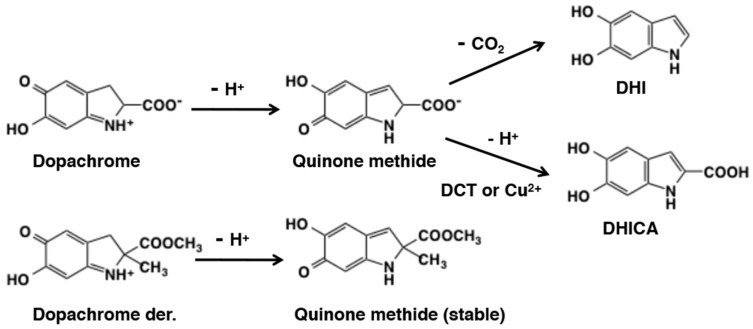
Reaction of dopachrome to produce 5,6-dihydroxyindole (DHI) and 5.6-dihydroxyindole-2-carboxylic acid (DHICA) and roles of dopachrome tautomerase (DCT) and Cu^2+^ ions [102,151,152,153,155].

**Figure 29 ijms-21-06080-f029:**
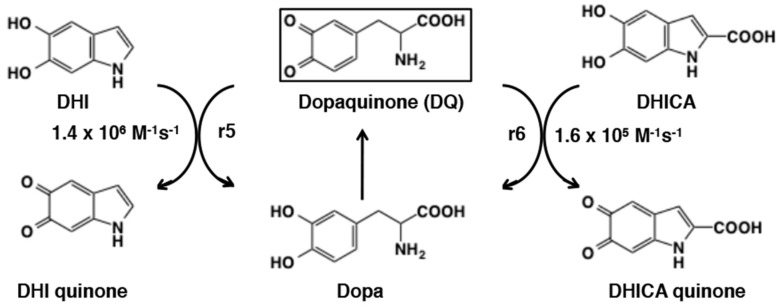
Kinetics of late stages of eumelanogenesis from DHI and DHICA. Note that dopaquinone (DQ) acts again here as a redox exchanger [156].

**Figure 30 ijms-21-06080-f030:**
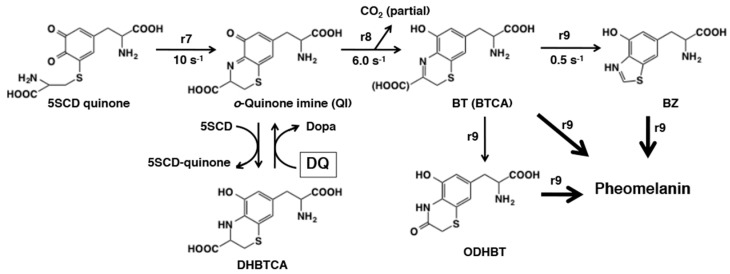
Kinetics of late stages of pheomelanogenesis from 5-*S*-cysteinyldopa (5SCD) quinone [166]. Note that dihydro-1,4-benzothiazine-3-carboxylic acid (DHBTCA) is produced *via* redox exchange while benzothiazine intermediates are generated by rearrangement of quinone imine intermediate (with/without decarboxylation). Rate constants are from Napolitano et al. [167]. Note that dopaquinone chemically controls pheomelanogenesis at a critical stage of DHBTCA oxidation.

**Figure 31 ijms-21-06080-f031:**
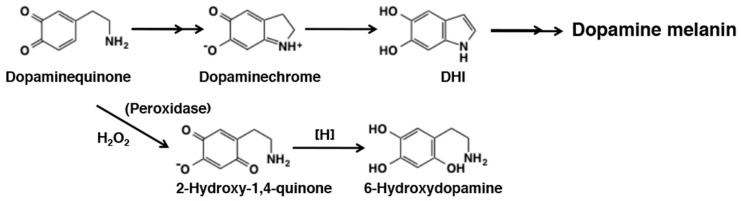
Reaction of dopaminequinone leading to the production of dopamine melanin through dopaminechrome [129,179]. Note that oxidation of dopamine with peroxidase/H_2_O_2_ gives 6-hydroxydopamine after reduction [67].

**Figure 32 ijms-21-06080-f032:**
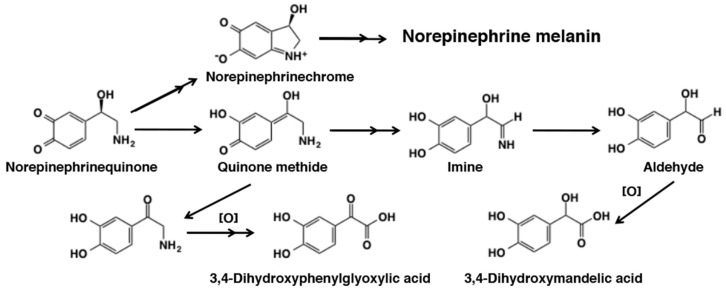
Reaction norepinephrinequinone [186]. The quinone methide pathway leads to the production of 3,4-dihydroxymandelic acid (DOMA), 3,4-dihydroxybenzaldehyde (DHBAld), 3,4-dihydroxybenzoic acid (DHBA), and 3,4-dihydroxyphenylglyoxylic acid. DHBAld and DHBA arise from DOMA (see Figure 17). Another pathway leads to the production of norepinephrine melanin through norepinephrinechrome.

**Figure 33 ijms-21-06080-f033:**
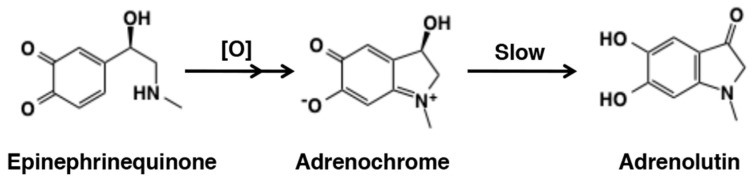
Reaction of epinephrinequinone [190]. The product adrenochrome gradually rearrange to adrenolutin (3,5,6-trihydroxy-*N*-methylindole).

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
