# Peer review of "Chemical Reactivities of ortho-Quinones Produced in Living Organisms: Fate of Quinonoid Products Formed by Tyrosinase and Phenoloxidase Action on Phenols and Catechols"

_ijms, 2020, doi:10.3390/ijms21176080_

Round 1
Reviewer 1 Report
This manuscript systematically discusses the chemical reaction of ortho-Quinones with some nucleophilic reagents, and explains the role in organisms, especially the mechanism of melanin formation. This is a high-level review article. However, some parts of the text should be revised
- The subject should be appropriately revised (for example: the chemical reactivity of o-quinine in living organisms. Phenol oxidation catalyzed by phenol and tyrosinase catechol). It will be more attractive to readers.
- p3 line 4 Suggest 3D protein structure of tyrosinase.
- p4 The tenth line at the bottom (…… into four categories (Figure 4): 1)…..to …Reaction 4) (very slow). It is cannot understand the relationship between 1) - 4) in the sentence and Figure 4, so please adjust the expression of the sentence
- P7 Fig. 8 The structure of the product in this equation is wrong (more CH2)
- P7 “3.3. Redox Exchange of o-Quinones with Reducing Agents” is changed to 3.6 in this paragraph. more suitable
- P9 Fig. 11 The ratio of 2 products is 5:1. Put one on the next line to make it clear
Author Response
We thank both the referees for their excellent suggestions and comments that will improve this manuscript dramatically. Point by point answers to referee’s queries are given below.
Answer to Referee’s Comments:
Reviewer 1.
1. The title should be revised.
Ans: The title is changed as: Chemical reactivities of ortho-quinones produced in living organisms: Fate of quinonoid products formed by tyrosinase and phenoloxidase action on phenols and catechols.
2. p3 line 4 Suggest 3D protein structure of tyrosinase.
Ans: We added the following sentence: The first 3D structure study of this protein indicates that three histidine residues each in the heavy chain coordinate to two copper atoms (CuA and CuB) [8].
3. p4 The tenth line at the bottom (…… into four categories (Figure 4): 1)…..to …Reaction 4) (very slow). It is cannot understand the relationship between 1) - 4) in the sentence and Figure 4, so please adjust the expression of the sentence.
Ans: We Changed these lines as follows:
With this exception in mind, the chemical reactivity of o-quinones can be classified into four categories based on the approximate speed of reactions (Figure 4): 1) addition of thiols (through sulfhydryl group) by 1,6-Michael reaction, 2) reduction to the parent catechols through redox exchange, 3) addition of amines by 1,4-Michael reaction or Schiff base formation, 4) addition of other nucleophiles such as carboxylic acids, phenols, alcohols and water. The rate of these four set of reactions are: Reaction 1 (fast)> Reaction 2 (next fast) >> Reaction 3 (slow)> Reaction 4 (very slow). In figure 4, the thickness of the arrows corresponds to the fastness of these reactions.
4. P7 Fig. 8 The structure of the product in this equation is wrong (more CH2)
Ans: Figure corrected.
5. P7 “3.3. Redox Exchange of o-Quinones with Reducing Agents” is changed to 3.6 in this paragraph. more suitable.
Ans. We made this change: Section 3.3 was changed to section 3.5 and moved before section 3.6. Section 3.4. (amine) became 3.3. and section 3. 5. (carboxyl) became section 3.4. Also at the end of this paragraph, it was changed to section 3.6.
6. P9 Fig. 11 The ratio of 2 products is 5:1. Put one on the next line to make it clear
Ans: We modified Fig.11 according to reviewer’s suggestion.
Reviewer 2 Report
The review, entitled “Chemical Reactivity of ortho-Quinones Produced by Tyrosinase-catalyzed Oxidation of Phenols and Catechols” summarizes the tyrosinase functions and the ortho-quinones (o-quinones) chemical reactivity, with special focus to mechanisms related to melanogenesis. The work is very interesting, and has been conceived to report the most recent knowledge about the o-quinones products in the biological systems, however, some problems, as indicated below, should be addressed before the document can be considered for the publication in this journal. This version of the manuscript is not enough complete.
Here, I present all my comments in detail, but my global consideration is almost positive.
- Minor comments:
The english language should be improved to ensure that an international audience can clearly understand the text. In fact, there are several points in which the syntax needs to be changed.
I suggest to add in the text the complete wording of the following terms (deoxy-, oxy-, met-, and deact-), specifically when the authors explain the different oxidation states of tyrosinase. Moreover, the authors indicate in italics the word "chemical" and the adverb “chemically". Why?
In general, I suggest to review the style of the manuscript according to the guidelines of the journal, and to standardize the indications shown on the figures.
Your introduction needs more details. I suggest that you improve the description about the general knowledge to provide more justification for your collect. The references in this section are not sufficient.The authors could add other references, such as:
- Judy L Bolton et al., 2018 -Formation and biological targets of botanical o-quinones.
- Shun Zhang et al., 2019 -Impact of Dopamine Oxidation on Dopaminergic Neurodegeneration.
- Anastasiia Y Glagoleva et al., 2020 -Melanin Pigment in Plants: Current Knowledge and Future Perspectives.
Introduction:
I suggest to explain better the goal of the review, and with this regard the authors aim their attention about tyrosinase functions, and the chemical reactivity of the o-quinones, produced in the melanogenesis mechanism, then I do not understand why non-enzymatic reactions are also considered.
In addition, the authors reported in the present review that -at this point we want to point out that practically all biological reactions are enzyme catalyzed, including such simple reaction as hydration of carbon dioxide to carbonic acid which can proceed quite rapidly without the need for an enzyme-
The authors should better explain this sentence, for example in the red cells, this simple reaction needs to anhydrase carbonic to proceed quite rapidly in order to form the final compound (Reinhart A.F. Reithmeier et al., 2016, Band 3, the human red cell chloride/bicarbonate anion exchanger (AE1,SLC4A1), in a structural context). In a biological system, without the enzyme, the reaction would occur very slowly to be useful without enzyme.
Conclusions:
The authors conclude that -o-quinones are able to modify the structure of proteins and DNA, leading possibly to cytotoxicity and carcinogenesis. Dopaquinone plays pivotal roles in melanogenesis; it acts chemically in several key steps including the redox exchange with cyclodopa to form dopachrome in the early stages of eumelanin production and the addition of cysteine to initiate pheomelanin production-
These findings are very interesting, the modification of proteins or DNA structures is an good aspects to investigate, in particular the cytotoxicity effects, consequently leading to carcinogenesis. o-Quinones are mostly produced by the oxidation of the corresponding catechols (o-diphenols) through tyrosinase enzyme. The high reactivity of thiol group renders proteins to react with o-quinones. This type of protein modifications is biologically important because this may lead to denaturation of thiol proteins and inhibition of thiol enzymes, leading to cytotoxicity.
Based on these evidence, are there studies in the literature in where proteins damages are shown, if yes, what are the specific proteins involved? For example proteins on the plasma membrane? In addition, the endogenous antioxidant system in these cells has been considered? What could be its role? And the diet supplementation of metabolites with antioxidant power, beyond ascorbic acid? I suggest to authors to add some references on these aspects, in order to make more complete the collection.
Author Response
We thank both the referees for their excellent suggestions and comments that will improve this manuscript dramatically. Point by point answers to referee’s queries are given below.
Answer to Referee’s Comments:
Reviewer 2.
Minor comments:
1. Suggestion 1. suggest to add in the text the complete wording of the following terms (deoxy-, oxy-, met-, and deact-), specifically when the authors explain the different oxidation states of tyrosinase.
Ans: These terms, which represent the different oxidation states of tyrosinase, have already been used as they are in previous papers, so we would like to use them as well.
2. Suggestion 2. the authors indicate in italicsthe word "chemical" and the adverb “chemically". Why?
Ans: We changed the words ‘chemical’ and ‘chemically’ in italics to normal words.
3. I suggest to review the style of the manuscript according to the guidelines of the journal, and to standardize the indications shown on the figures.
Ans: I unified the style of the manuscript according to the guidelines of the journal, and standardized the indications shown on the figures.
4. Introduction:
Introduction needs more details an should include some more references
Ans: These new references are all added, thus, reference numbers were changed in order. The introduction is expanded.
5. I suggest to explain better the goal of the review, and with this regard the authors aim their attention about tyrosinase functions, and the chemical reactivity of the o-quinones, produced in the melanogenesis mechanism, then I do not understand why non-enzymatic reactions are also considered.
In addition, the authors reported in the present review that -at this point we want to point out that practically all biological reactions are enzyme catalyzed, including such simple reaction as hydration of carbon dioxide to carbonic acid which can proceed quite rapidly without the need for an enzyme-
The authors should better explain this sentence, for example in the red cells, this simple reaction needs to anhydrase carbonic to proceed quite rapidly in order to form the final compound (Reinhart A.F. Reithmeier et al., 2016, Band 3, the human red cell chloride/bicarbonate anion exchanger (AE1,SLC4A1), in a structural context). In a biological system, without the enzyme, the reaction would occur very slowly to be useful without enzyme.
The lines 64-68 – “At this point we want to point out that practically all biological reactions are enzyme catalyzed, including such simple reaction as hydration of carbon dioxide to carbonic acid which can proceed quite rapidly without the need for an enzyme. However, the reactions of quinones form a group of nonenzymatic reactions that often essential for biological systems. In other words, most of the reactions of o-quinones can occur solely as chemical reactions without any assistance from enzymes.
Ans: the lines 64-68 were written as follows:
At this point we want to point out that practically all biological reactions are enzyme catalyzed, with the exception of ribozymes, which are RNA catalysts. Without enzymatic intervention, some reactions can proceed, but for them to be useful for biological system, enzymatic intervention is absolutely necessary. An example is hydration of carbon dioxide to carbonic acid which can proceed quite rapidly without the need for an enzyme, but in biological systems, carbonic anhydrase makes this hydration go faster. However, a very small set of reactions, form a group of nonenzymatic reactions that are often occur in biological systems. Glycation of hemoglobin is one such example. While it is a useful index to determine the seriousness of diabetic conditions, the glycation is entirely of nonenzymatic origin. In the case of o-quinone chemistry, once these reactive intermediates are generated by enzyme assisted reactions, the rest of the reactions seem to proceed nonenzymatically without the need for any enzymatic assistance. These nonenzymatic reactions are absolutely essential for parts of melanin biosynthesis, insect cuticular sclerotization, innate immunity in invertebrate animals where encapsulation and melanization of foreign objects occur, mussel glue formation (1, 4-6). Therefore, it is important to investigate the fate of o-quinones in biological systems, which forms the main goal of this review.
6. Conclusion: are there studies in the literature in where proteins damages are shown, if yes, what are the specific proteins involved? For example proteins on the plasma membrane? In addition, the endogenous antioxidant system in these cells has been considered? What could be its role? And the diet supplementation of metabolites with antioxidant power, beyond ascorbic acid? I suggest to authors to add some references on these aspects, in order to make more complete the collection.
Ans: Protein damage is widely accepted in the phenoloxidase literature. The oxidative browning of the potato, mushroom, eggplant, etc., cause not only the loss of reducing power in the cell, but also by the reactions of o-quinones with proteins (and perhaps nucleic acids). Studying individual reactions and showing what reaction is exactly happening is often very difficult. So many scientists use model studies. For instance, Vithayathil et al., (ref 39-42) showed that o-quinone can react with RNAse and modify its methionine. Our group has shown the reactions of cysteine with quinones. Protein polymerization with a number of o-quinones have been well studied and demonstrated [one example - Tyrosinase catalyzed protein polymerization as an in vitro model for sclerotization of insect cuticle. M. Sugumaran, B. Hennigan, J. O’Brien. Arch. Insect Biochem. Physiol. 6, 9-25 (1987)]. In real system, no one has deeply looked into which protein gets modified in living cells. It is very hard to study this reaction. The endogenous antioxidant system will quench the quinones as long as they are present. Once they are depleted, quinones will start reaction with cellular nucleophiles. Diet supplements of antioxidants can help only to certain extent. They are not uncontrollably taken by any cell to be helpful to quench all the reactions of quinones. Addressing these aspects is beyond the scope of this review. That is why such matters were not covered in this review.